# PDEfuncta: Spectrally-Aware Neural Representation for PDE Solution Modeling

**Minju Jo**[1*]   **Woojin Cho**[2*]   **Uvini Balasuriya Mudiyanselage**[3]
**Seungjun Lee**[4]   **Noseong Park**[5]   **Kookjin Lee**[3†]

[1]Sorbonne University,   [2]TelePIX,   [3]SCAI, Arizona State University,
[4]Brookhaven National Laboratory,   [5]KAIST
`minnju42@gmail.com, woojin.py@gmail.com, ubalasur@asu.edu,`
`slee19@bnl.gov, noseong@kaist.ac.kr, kookjin.lee@asu.edu`

## Abstract

Scientific machine learning often involves representing complex solution fields that exhibit high-frequency features such as sharp transitions, fine-scale oscillations, and localized structures. While implicit neural representations (INRs) have shown promise for continuous function modeling, capturing such high-frequency behavior remains a challenge—especially when modeling multiple solution fields with a shared network. Prior work addressing spectral bias in INRs has primarily focused on single-instance settings, limiting scalability and generalization. In this work, we propose Global Fourier Modulation (GFM), a novel modulation technique that injects high-frequency information at each layer of the INR through Fourier-based reparameterization. This enables compact and accurate representation of multiple solution fields using low-dimensional latent vectors. Building upon GFM, we introduce PDEfuncta, a meta-learning framework designed to learn multi-modal solution fields and support generalization to new tasks. Through empirical studies on diverse scientific problems, we demonstrate that our method not only improves representational quality but also shows potential for forward and inverse inference tasks without the need for retraining.

## 1 Introduction

Solving partial differential equations (PDEs) is essential to progress in many science and engineering applications, including fluid dynamics, seismic inversion, climate modeling, and mechanical design. Traditionally, these PDE problems are addressed using numerical methods (e.g. finite-difference [1], finite-volume [2], finite-element [3]), which are reliable but relatively slow and computationally expensive for high-dimensional or high-resolution data. Scientific Machine Learning (SciML) [4, 5, 6, 7, 8] offers a more efficient alternative by combining physics-based models with data-driven approaches to approximate solutions quickly and accurately. One critical challenge within SciML applications involves the efficient compression and functional representation of large-scale scientific datasets, which contain complex structures, sharp transitions, and high-frequency content [9, 10]. To handle such data is critical — not only for reducing storage and memory costs but also for enabling fast and flexible adaptation in downstream tasks such as operator learning, inverse modeling.

A promising strategy to address these challenges is the use of implicit neural representations (INRs), which represent data as continuous functions by mapping coordinates $\mathcal{X}$ (e.g. spatial, temporal

---

[*]Equal Contribution

[†]Corresponding Author

39th Conference on Neural Information Processing Systems (NeurIPS 2025).

positions) to physical quantities $f(\mathcal{X})$. These methods inherently offer mesh-free and resolution-invariant representations, making them particularly suitable for complex scientific data. Nevertheless, standard INR architectures based on multilayer perceptrons (MLPs) suffer from a well-known limitation called spectral bias [11, 12, 13], where the neural network tends to learn low-frequency components more rapidly than high-frequency ones.

This problem has been rigorously formalized in prior theoretical work. In particular, according to Theorem 1 in [11], MLPs with `tanh` activation, gradients associated with high-frequency components are smaller than those for low-frequency components.

Although various approaches, including SIREN [14], Fourier feature networks (FFN) [15], and wavelet implicit neural representations (WIRE) [16], have demonstrated improved performance, mitigating this spectral bias problem, by proposing advanced nonlinear activation functions, these methods typically focus on training a single INR for a single data instance.

To resolve this weakness, we turn to the Functa paradigm [17], which functionalizes discrete data—representing each sample as a continuous function and compresses an entire dataset by modulating a single INR with low-dimensional latent vectors. Motivated by this success, we ask whether a similar strategy can functionalize discretized PDE data—thereby collapsing massive simulation outputs into lightweight latents while restoring their functional character and use this reduced representation in performing downstream tasks (e.g., super-resolution, learning

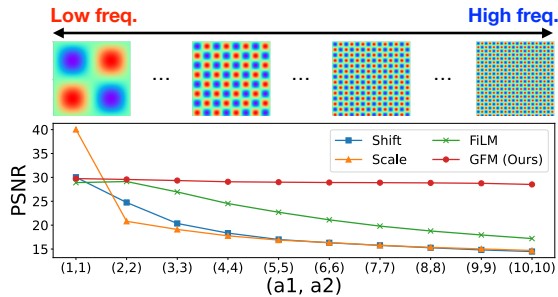

Figure 1: [Spectral bias problem] Experimental results on Helmholtz equations

forward/inverse PDE operators). However, naïvely adopting the Functa paradigm to the PDE domain is expected to bring novel challenges, namely, spectral bias and no established mechanism describing forward/inverse problems. See Figure 1 for example; existing INR modulation techniques (e.g., Shift, Scale, FiLM) struggle to capture high-frequency components of the PDE solution.

**Spectral bias of INR with modulation methods.** Existing modulation approaches [17, 18] apply instance-specific modulations directly in the latent space, without explicit control over spectral characteristics. Consequently, these modulation methods inadequately represent PDE data with complex, high-frequency features. To effectively overcome spectral bias within modulated INR settings, we propose a novel modulation method based on Fourier reparameterization, a frequency-aware method that explicitly injects spectral priors into the latent space.

In this paper, we address these challenges by introducing a spectrally balanced and resolution-invariant framework for learning scientific data. We make the following key contributions:

- We propose Global Fourier Modulation (GFM), a modulation technique that extends Fourier reparameterization to the modulated INR setting. Our GFM injects fixed Fourier bases into each layer of the shared INR backbone, enabling it to learn high-frequency components that other modulations fail to capture.

- We present PDEfuncta, a reversible architecture that represents input-output solution field pairs using one shared latent modulation vector. This design supports bidirectional inference.

- Through various experiments on benchmark PDE datasets, we empirically demonstrate that our method achieves high reconstruction accuracy, generalizes across parameter ranges and resolutions, and outperforms existing modulation baselines.

Our approach also bridges the gap between INR-based compression and operator learning. While classical neural operators [19, 20, 21, 22, 23] are designed to map between function spaces, they are limited to one-way inference. In contrast, our PDEfuncta leverages the functional continuity and Fourier reparameterization based modulation to deliver a compact, flexible, and bidirectional inference for learning function space mappings in the latent space.

## 2 Related Works

**Implicit Neural Representation** INR methodologies aim to receive coordinates as input and map them to corresponding values. Two types of INR, namely Physics-informed Neural Network (PINN) [24] and Neural Radiance Fields (NeRF) [25], have the goal of solving PDE problems and representing 3D scenes, respectively. Since inference is possible through input coordinates, it is feasible to represent at the desired resolution and express a single data point as a function. According to [15], Fourier feature mapping can maximize the performance of INR, showing equivalence to the shift-invariant kernel method from the perspective of NTK [26]. On the other hand, as seen in [14, 16, 27], many studies have focused on proposing activation functions for INR to enhance both the expressiveness and complexity of the INR model.

**Data to Functa** An INR is a network that continuously represents a signal or field in its domain. INR methods like DeepSDF [28] and NeRF model data as continuous functions. Recent work has explored treating data as functions to improve learning in a function space. For example, Functa [17] shows that one can encode each data sample like an image, as a function (parameters of an INR) and learn tasks over these representations. In this paradigm, each dataset sample is encoded as a low-dimensional latent vector that modulates a shared INR network. Such functional representations offer substantial benefits, including resolution independence, improved generalization, and efficient data compression. Our work leverages these ideas by employing modulated INRs to encode PDE solution fields and enable direct learning of operators in continuous function spaces, significantly improving efficiency and flexibility in scientific machine learning applications.

**Neural Operator** Neural operator is one of the methods to approximate solutions to PDEs, aiming to map input and output function spaces. Fundamental research in this field, known as the deep operator network (DeepONet) [29], is composed of a branch network and a trunk network, each used to learn the PDE operator. In this process, DeepONet requires a fixed discretization of the input function. On the other hand, there are studies that learn operator by approximating kernel integral operations. In this domain, a promising study known as Fourier neural operator (FNO) [20] maps function spaces using convolution operation in the Fourier domain. Furthermore, investigations involve training models in the wavelet domain [30, 31] and utilizing message passing in graph neural networks for operator learning [32, 33]. These approaches are highly effective in learning PDE data and can perform computations rapidly during dynamic simulations. However, a drawback of these methods is the requirement for input and output data to have consistent sampling point. Subsequently, solutions such as the [34, 35], which feature a GNN-based encoder-decoder structure, have been proposed to address problems beyond fixed rectangular domains. Additionally, there are DINO [22] and CORAL [36] based on implicit neural representation, which enables learning and inference even in situations where the sampling ratios of input and output data differ.

## 3 Global Fourier Modulation for Mitigating the Spectral Bias Problem

INRs model data as continuous functions $f(\mathcal{X}; \theta)$ that map spatio-temporal coordinates $\mathcal{X}$ to solution fields, offering resolution-invariant representations. However, they require a separately trained set of parameters $\theta = \{W, b\}$ for each data sample, which limits scalability. The Functa framework [17, 37] addresses this scalability issue by introducing modulated INRs, where a single shared INR network is used across all samples, and each instance is customized via a sample-specific latent code $z$. This latent vector is passed through a lightweight modulation network $g(z; \pi)$ to produce a set of layer-wise modulation vectors $\{\alpha^k\}_{k=1}^L$, where $\pi = \{W_{\text{mod}}, b_{\text{mod}}\}$ are the parameters of $g$. The latent vector $z$ is learnable and initialized as a zero vector, forming a set known as *Functaset*.

However, modulated INRs still inherit the spectral bias of MLPs, where low-frequency components are learned more rapidly than high-frequency ones. To mitigate this, we propose Global Fourier Modulation (GFM), which is a novel modulation strategy that explicitly injects frequency-awareness into the network. Its design and theoretical foundation are detailed next.

### 3.1 Spectral Bias and Fourier Reparameterization

INRs are well known for exhibiting spectral bias, the tendency of neural networks to approximate the low-frequency components of a target function much earlier than the high-frequency components.

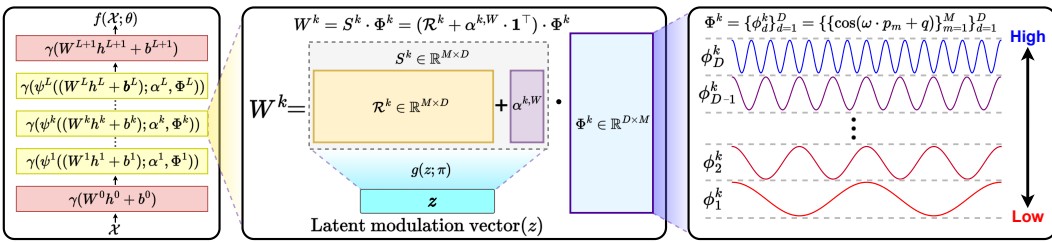

Figure 2: **Overall pipeline of GFM.** The GFM framework injects fixed Fourier bases into each layer of the shared INR, enabling effective learning of both low- and high-frequency components.

Theorem 1 from [11] provides a theoretical explanation for this phenomenon, showing that low-frequency components dominate the gradient dynamics near initialization. As a result, model struggle to capture fine-scale structures, which are essential in many physical systems and PDE-based data.

To mitigate this issue, prior works has proposed frequency-aware parameterizations that explicitly embed spectral priors into the model architecture [38]. A representative approach is to reparameterize each weight matrix as $W = S \cdot \Phi$, where $S \in \mathbb{R}^{M \times D}$ is a trainable coefficient matrix and $\Phi \in \mathbb{R}^{D \times M}$ is a fixed Fourier basis composed of sinusoidal functions. This formulation restricts the function class to lie within a predefined frequency subspace, making high-frequency modes more accessible during training. It also improves the conditioning of the Neural Tangent Kernel (NTK), leading to a more balanced eigenvalue spectrum and better convergence on complex signals.

**Theorem 1** *(Theorem 2 in [38]) Let $W \in \mathbb{R}^{M \times M}$ denote the weight matrix of a hidden layer in an MLP. Suppose that it is reparameterized as $W = S \cdot \Phi$, where $S \in \mathbb{R}^{M \times D}$ is a trainable coefficient matrix and $\Phi \in \mathbb{R}^{D \times M}$ is a fixed Fourier basis matrix. Then, for any two frequencies $\mathcal{F}1 > \mathcal{F}2 > 0$, any $\epsilon \geq 0$, and fixed index $i$, for $m = 1, \ldots, M$ there must exist a set of basis matrices such that:*

$$\left| \frac{\partial \mathbb{L}(\mathcal{F}_1)}{\partial s_{i,m}} \middle/ \frac{\partial \mathbb{L}(\mathcal{F}_2)}{\partial s_{i,m}} \right| \geq max \left\{ \left| \frac{\partial \mathbb{L}(\mathcal{F}_1)}{\partial w_{i,1}} \middle/ \frac{\partial \mathbb{L}(\mathcal{F}_2)}{\partial w_{i,1}} \right|, ..., \left| \frac{\partial \mathbb{L}(\mathcal{F}_1)}{\partial w_{i,M}} \middle/ \frac{\partial \mathbb{L}(\mathcal{F}_2)}{\partial w_{i,M}} \right| \right\} - \epsilon, \tag{1}$$

*where $s_{i,m}$ and $w_{i,m}$ denote elements of $S$ and $W$, respectively.*

Theorem 1 formalizes the gradient dynamics under Fourier reparameterization. It shows that the gradient ratio between high-frequency and low-frequency components is greater under Fourier reparameterization than standard weights, up to a small error $\epsilon$. This indicates that high-frequency structures receive relatively stronger gradient signals during training, improving the model's ability to learn fine-scale patterns.

**Construction of the Fourier Basis.** The fixed basis matrix $\Phi^k \in \mathbb{R}^{D \times M}$ is constructed by evaluating phase-shifted cosine functions over a uniform spatial grid $\{p_m\}_{m=1}^{M}$ in the range $[-T_{\max}/2, T_{\max}/2]$, where $T_{\max}$ is determined by the smallest frequency used. Each row corresponds to a unique combination of frequency $\omega$ and phase shift $q$, allowing spectral diversity across layers. Specifically, each basis row in the $k$-th layer is defined as:

$$\phi_d^k = \{\cos(\omega \cdot p_m + q)\}_{m=1}^{M}, \quad \text{for } d = 1, \ldots, D \tag{2}$$

where $\omega$ is selected from a mixture of low and high frequency components, and $q$ is a phase shift uniformly sampled from $[0, 2\pi]$. This diverse basis configuration enables the network to better capture both smooth and high-frequency features. Full construction details are provided in Appendix H.

### 3.2 Global Fourier Modulation for Modulated INRs

While Fourier reparameterization [38] effectively mitigates spectral bias in standard INRs by projecting weights into a frequency-aware subspace, it is fundamentally restricted to single-instance settings. In contrast, modulated INRs aim to generalize across datasets by injecting instance-specific information through modulation. However, prior modulation techniques—such as Shift, Scale, and FiLM—operate solely in the activation space and lack explicit frequency control. This limitation makes them inadequate for modeling signals with rich high-frequency content (see Figure 6).

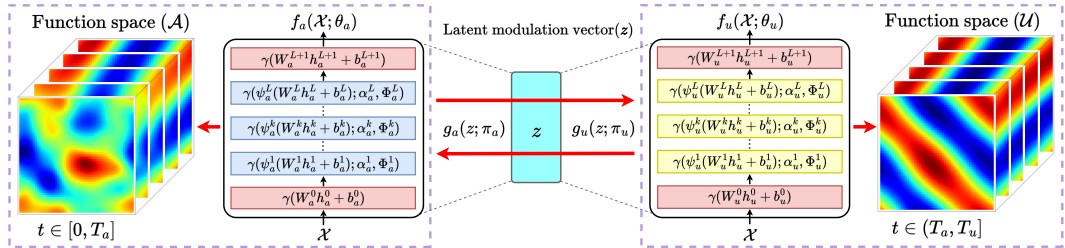

Figure 3: **Proposed method: PDEfuncta.** PDEfuncta leverages GFM to represent paired function spaces using a shared latent vector, supporting bidirectional inference between input and output fields. This architecture enables efficient compression and reversible mapping for scientific datasets.

To address this gap, we propose GFM, a novel modulation strategy that integrates frequency-awareness directly into the weight space. GFM extends Fourier reparameterization to multi-sample settings by combining a shared Fourier basis $\Phi^k$ (cf. Section 3.1) with sample-specific modulation vector $\alpha^k = \{\alpha^{k,W}, \alpha^{k,b}\}$. Here, $\alpha^{k,W} \in \mathbb{R}^d$ modulates the spectral components of the weight matrix, while $\alpha^{k,b} \in \mathbb{R}^d$ adjusts the bias term. At each layer $k$, the weight matrix is computed as:

$$W^k = S^k \cdot \Phi^k = (\mathcal{R}^k + \alpha^{k,W} \cdot \mathbf{1}^\top) \cdot \Phi^k, \tag{3}$$

where $\mathcal{R}^k \in \mathbb{R}^{d \times M}$ is a shared learnable base matrix and $\mathbf{1} \in \mathbb{R}^M$ is a vector of ones for broadcasting. This construction enables direct, sample-specific modulation in the frequency domain. To clarify how GFM differs from previous approaches, we first revisit the general update rule for modulated INRs:

$$h^{k+1} = \gamma \left( \psi(W^k \cdot h^k + b^k; \alpha^k) \right), \tag{4}$$

where $h^k \in \mathbb{R}^d$ is the hidden representation at layer $k$, $\gamma$ is an activation function and $\psi(\cdot; \alpha^k)$ is a modulation function. Existing modulation methods apply $\psi$ in the activation space as follows:

$$\text{Shift} : \psi((W^k \cdot h^k + b^k); \alpha^k) = (W^k \cdot h^k + b^k) + \alpha^k, \tag{5}$$

$$\text{Scale} : \psi((W^k \cdot h^k + b^k); \alpha^k) = (W^k \cdot h^k + b^k) \odot \alpha^k, \tag{6}$$

$$\text{FiLM} : \psi((W^k \cdot h^k + b^k); \alpha^k) = (W^k \cdot h^k + b^k) \odot \alpha^{k,W} + \alpha^{k,b}. \tag{7}$$

These approaches adjust values per instance but leave the weight matrices unchanged, limiting their capacity to modulate frequency content explicitly. In contrast, GFM redefines the transformation as:

$$\text{GFM (Ours)} : \psi((W^k \cdot h^k + b^k); \alpha^k, \Phi^k) = (\mathcal{R}^k + \alpha^{k,W} \cdot \mathbf{1}^\top) \cdot \Phi^k \cdot h^k + b^k + \alpha^{k,b}, \tag{8}$$

While the primary effect arises from spectral modulation via $\alpha^{k,W}$, the additional bias term $\alpha^{k,b}$ enhances representational flexibility and fine-grained control over output activations.

By conditioning weights on a shared Fourier basis and modulating them in the frequency domain, GFM inherits the benefits of Fourier reparameterization—such as improved NTK conditioning and stronger high-frequency response (Theorem 1)—while scaling to dataset-level representations. This design enables frequency-targeted, sample-specific adaptation in a parameter-efficient manner.

## 4  PDEfuncta: From discrete to continuous solution representations

Applying our proposed GFM to the Functa framework effectively overcomes the spectral bias compared to conventional modulation strategies (see Table 1 and Figure 6 in Section 5). As demonstrated theoretically in Section 3, GFM significantly improves the compression and accurate reconstruction of scientific data with high-frequency components. While effectively modeling individual datasets is valuable, scientific applications frequently involve data pairs defined on two distinct yet related function spaces $(\mathcal{A}, \mathcal{U})$. A representative example is full waveform inversion (FWI) [39, 10], where paired data samples exist as seismic velocity fields and corresponding seismic waveforms (cf. Appendix D). In this section, leveraging Functa's functionalization and compression capabilities, we introduce PDEfuncta, a novel framework designed to jointly represent these paired function spaces through a shared latent modulation vector $z$.

## 4.1 Overall Architecture

Our proposed method, as depicted in Figure 3, represents functions in the function space $\mathcal{A}$ and function space $\mathcal{U}$ through respective neural networks $f_a(\mathcal{X}; \theta_a)$ and $f_u(\mathcal{X}; \theta_u)$. These two INR models receive spatio-temporal coordinates $h^0 = [x, t]^\top$ as input and have the following structure:

$$
\begin{aligned}
h^1 &= \gamma(W^0 \cdot h^0 + b^0), \\
h^{k+1} &= \gamma(\psi^k(W^k \cdot h^k + b^k); \alpha^k, \Phi^k), \quad k = 1, \ldots, L, \\
f(\mathcal{X}; \theta) &= W^{L+1} \cdot h^{L+1} + b^{L+1}.
\end{aligned}
\tag{9}
$$

Here, $\theta_a = \{W_a^k, b_a^k\}_{k=1}^L$ and $\theta_u = \{W_u^k, b_u^k\}_{k=1}^L$ are the weight and bias of the INRs $f_a$ and $f_u$, respectively, which are common model parameters used for samples in the dataset. In contrast, $\alpha^k$ represents the modulation vector for the $k$-th layer, serving as model parameters that vary depending on the data as follows:

$$
\begin{aligned}
\{\alpha_a^k\}_{k=1}^L &= g_a(z; \pi_a), \quad g_a(z; \pi_a) = W_a^{\mathrm{mod}} z + b_a^{\mathrm{mod}} \\
\{\alpha_u^k\}_{k=1}^L &= g_u(z; \pi_u), \quad g_u(z; \pi_u) = W_u^{\mathrm{mod}} z + b_u^{\mathrm{mod}}.
\end{aligned}
\tag{10}
$$

The latent modulation vector $z$ consists of parameters learned through auto-decoding [28] from data sample, and one $z$ is generated for each data sample. The weights and bias parameters constituting a linear layers, $\pi_a = [W_a^{\mathrm{mod}}, b_a^{\mathrm{mod}}]$, $\pi_u = [W_u^{\mathrm{mod}}, b_u^{\mathrm{mod}}]$, receive the $z$ as input and output modulation vectors $\{\alpha_a^k, \alpha_u^k\}_{k=1}^L$, (i.e. Eq (10)), these two modulation vectors are applied to each layer of the INR networks $f_a$ and $f_u$, enabling the neural network to represent various data samples. We extend Fourier reparameterization to support multiple data instances via GFM.

$$
W_a^k(\alpha) = S^k \cdot \Phi^k = (\mathcal{R}_a^k + \alpha_a^{k,W} \cdot \mathbf{1}^\top) \cdot \Phi^k, \quad W_u^k(\alpha) = S^k \cdot \Phi^k = (\mathcal{R}_u^k + \alpha_u^{k,W} \cdot \mathbf{1}^\top) \cdot \Phi^k, \tag{11}
$$

where $\{\mathcal{R}_a^k, \mathcal{R}_u^k\}_{k=1}^L$ is a base coefficient matrix (shared across all instances) and $\{\alpha_a^{k,W}, \alpha_u^{k,W}\}_{k=1}^L$ are a modulation term derived from a shared latent modulation vector $\mathbf{z}$. In the simplest instantiation, the GFM method adds $\{\alpha_a^k, \alpha_u^k\} \in \mathbb{R}^D$ to each row of $\{\mathcal{R}_a^k, \mathcal{R}_u^k\}_{k=1}^L \in \mathbb{R}^{M \times D}$ (or an equivalent broadcasting) so that $\{\alpha_a^k, \alpha_u^k\}$ adjust the coefficients associated with the $D$ Fourier basis vectors.

## 4.2 How to train

Our training method requires steps to learn the functaset that matches to train/testset during both the training and testing phases. We use a nested loop structure in the training phase (cf. Algorithm 1) following functa framework: an inner loop that focuses on construction of functaset, and an outer loop that updates the global model parameters. Both loops use the same loss function $\mathbb{L}$, which is the sum of reconstruction errors from $f_a$ and $f_u$ as follows:

$$
\mathbb{L}_i = (\mathbb{L}_{\mathcal{X}_i}(a^i, f_a(\mathcal{X}; \theta_a, g_a(z^i; \pi_a))) + \mathbb{L}_{\mathcal{X}_i}(u^i, f_u(\mathcal{X}; \theta_u, g_u(z^i; \pi_u)))). \tag{12}
$$

While both loops utilize the same loss function $\mathbb{L}_{\mathcal{X}}(x, \hat{x}) = (x - \hat{x})^2$, the inner loop updates parameters for each individual sample, whereas the outer loop updates the parameters based on the batch loss $\sum_{i=1}^{\mathcal{B}} \mathbb{L}_i$. In the inner loop, we sample $K$ samples from each batch and update modulations of the selected samples. This bilevel approach mimics a specialized form of Model-Agnostic Meta-Learning (MAML) [40] that focuses on learning only a subset of weights, facilitating efficient adaptation and learning across diverse datasets [41, 17].

# 5 Experiments

In this section, we empirically evaluate the proposed GFM and PDEfuncta frameworks. Our experiments focus on four main goals: (i) validating GFM's ability to compress and reconstruct PDE solution fields under single-INR modulation; (ii) evaluating whether the learned latent space enables generalization to unseen parameter configurations; (iii) demonstrating PDEfuncta's ability to perform bidirectional inference across paired function spaces; and (iv) comparing PDEfuncta against neural operator baselines on complex geometry benchmarks.

Table 1: Comparison with existing modulation methods (PSNR↑/MSE↓)

| Modulation | Convection | | Helmholtz #1 | | Helmholtz #2 | | Navier-Stokes | | Kuramoto-Sivashinsky | |
|---|---|---|---|---|---|---|---|---|---|---|
| | PSNR | MSE | PSNR | MSE | PSNR | MSE | PSNR | MSE | PSNR | MSE |
| Shift [17] | 13.225 | 0.3849 | 25.467 | 0.0341 | 18.249 | 0.1426 | 25.767 | 0.0092 | 16.962 | 7.9715 |
| Scale [17] | 27.410 | 0.0067 | 25.669 | 0.0472 | 18.183 | 0.1355 | 31.469 | 0.0024 | 20.395 | 3.8306 |
| FiLM [22] | 27.330 | 0.0125 | 33.611 | 0.0133 | 22.771 | 0.0762 | 31.904 | 0.0022 | 22.297 | 2.3309 |
| GFM | **42.474** | **0.0004** | **39.139** | **0.0006** | **29.033** | **0.0056** | **34.754** | **0.0011** | **29.827** | **0.4198** |

## 5.1 Experimental Setup

For each dataset, we report reconstruction performance using Peak Signal-to-Noise Ratio (PSNR) and Mean Squared Error (MSE). For neural operator tasks, we use the $L^2$-norm of prediction error, which is the standard metric in literature. All modulation-based experiments use a 20-dimensional latent vector $\mathbf{z}$ to modulate the shared INR. Further details are provided in Appendix E.

**Datasets.** We evaluate the performance of our proposed GFM using four scientific datasets: the convection equation (Eq. (13)), Helmholtz equation (Eq. (14)), Navier–Stokes (NS) equation (Eq. (17)), and Kuramoto–Sivashinsky (KS) equation (Eq. (16)). To assess capability of PDEfuncta for bidirectional inference, we use the OpenFWI dataset [10], which contains pairs of samples across two distinct function spaces. Lastly, we explore performance of our proposed method in a challenging neural-operator scenario, using the Airfoil (Eq. (21)) and Pipe flow (Eq. (22)) datasets. Detailed descriptions of each dataset are provided in Appendix D.

**Baselines.** We compare our GFM against three representative modulation methods: Shift, Scale, and FiLM. Additionally, we provide comparisons with Spatial Functa [37] in Appendix J. To evaluate the neural operator capability of PDEfuncta, we consider established baselines including FNO [20], UNet [42], Geo-FNO [35], FFNO [21], CORAL [36] and MARBLE [23].

## 5.2 Compression and Reconstruction with Single-INR modulation

We begin with the standard single-INR setting, where a single INR is modulated by a sample-specific latent codes to compress and reconstruct individual continuous solution fields. Experiments are conducted on four PDE datasets known for high-frequency details: 1D convection equations ($\beta \in \{1, \ldots, 50\}$), 2D Helmholtz equations (#1: $a_1, a_2 \in \{1, \ldots, 5\}$; #2: $a_1, a_2 \in \{1, \ldots, 10\}$), 2D incompressible NS, and KS equations. Detailed experimental setup is listed in Appendix E.

Figure 4: [Kuramoto–Sivashinsky] Training loss (MSE) over epochs.

As summarized in Table 1, GFM achieves the highest PSNR and lowest MSE across all datasets. Notably, it improves PSNR by up to 15dB over the strongest baseline, FiLM, on the convection equation, and by 3-7dB on the others. Figure 4 shows the training loss curves up to 5000 epochs on the KS equation for four modulation methods: Shift, Scale, FiLM, and our proposed GFM. GFM shows the fastest and most stable convergence, reaching the lowest final MSE among all baselines. This also highlights that GFM can successfully compress and reconstruct high-frequency spatiotemporal dynamics using only a 20-dimensional latent code. We also present qualitative reconstruction results in Figure 6. The top row shows KS error maps, and the bottom row shows Helmholtz reconstructions. GFM shows the most visually accurate outputs with minimal error, while others show noticeable distortion.

### 5.2.1 Evaluating Functional Representation in Latent Modulation Space

To evaluate whether the learned latent modulation space captures meaningful functional representations, we design two generalization experiments. These experiments test the model's ability to interpolate beyond the discrete set of PDE coefficients seen during training. We report results on the convection equation, where we test on unseen coefficients $\beta^* \in \{1.5, 2.5, \ldots, 49.5\}$ that are excluded from the training set ($\beta \in \{1, 2, \ldots, 50\}$). In both cases, we aim to recover the full solution for $\beta^*$ using

Table 2: Results on unseen coefficients

| Method | Setting 1 | | Setting 2 | |
|---|---|---|---|---|
| | PSNR | MSE | PSNR | MSE |
| Shift | 8.427 | 0.5305 | 13.307 | 0.4533 |
| Scale | 9.125 | 0.4186 | 9.197 | 0.5421 |
| FiLM | 10.613 | 0.3954 | 11.927 | 0.3268 |
| GFM | **32.586** | **0.0017** | **30.830** | **0.0033** |

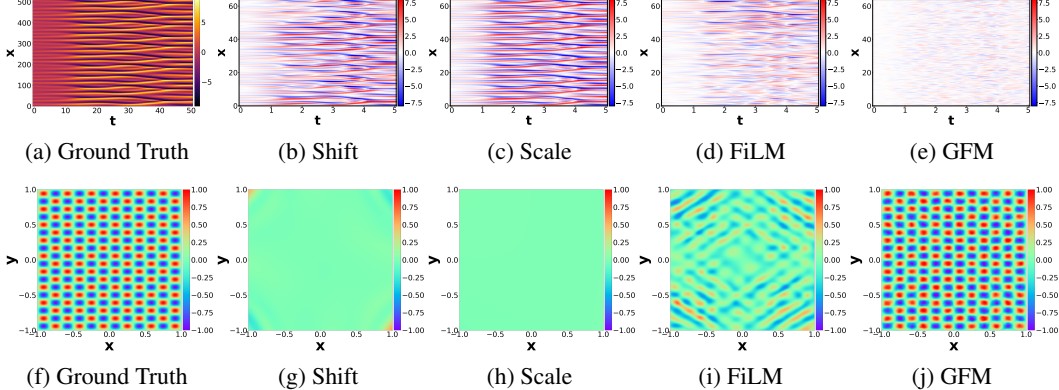

Figure 6: Top: Error maps for the Kuramoto–Sivashinsky equation. Bottom: Helmholtz equation ($a_1 = 6.0, a_2 = 9.0$) reconstructions. GFM achieves the most accurate reconstructions.

only its inferred latent code $z$, without updating the shared INR. All results are obtained using the pre-trained model from Section 5.2. Additional details are provided in Appendix F.2.

**Setting 1: Latent Interpolation.**     We assume no access to the target solution for the unseen coefficient $\beta^*$. Its latent code $z_{\beta^*}$ is estimated via cubic interpolation between the latent codes of nearby training coefficients. This setting evaluates whether the latent space encodes smooth, functionally meaningful transitions between parameter-conditioned solutions.

**Setting 2: Latent Fitting from Partial Observation.**     We assume partial access to the target solution over $t \in [0, 0.5]$ and optimize only the latent code $z_{\beta^*}$ to fit the observed segment while freezing the shared INR. We evaluate reconstruction performance on both the observed segment and the unobserved future $t \in [0.5, 1]$.

Our experimental results, which are summarized in Table 2 and Figure 5, empirically demonstrate that, even when trained only on a finite discrete set of PDE coefficients, the proposed method can effectively generalize to unseen continuous parameter values via latent space interpolation. This indicates that the latent modulation vectors capture a smooth, functional mapping aligned with the underlying parametric dependence of the PDE family. In Setting 2, where partial observations of the solution are available, our method successfully recovers the full spatiotemporal structure of the solution by optimizing only the latent code while keeping the INR fixed. The

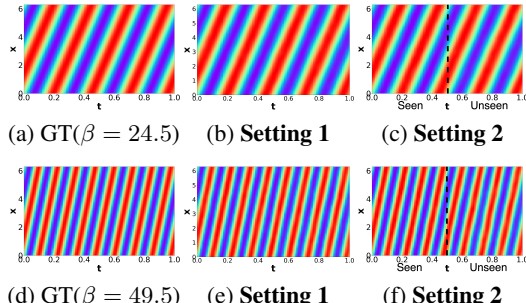

(a) GT($\beta = 24.5$)    (b) **Setting 1**    (c) **Setting 2**

(d) GT($\beta = 49.5$)    (e) **Setting 1**    (f) **Setting 2**

Figure 5: [Convection] Reconstruction results for unseen $\beta = 24.5, 49.5$ on Setting 1 and 2.

model not only fits the observed interval accurately but also extrapolates reliably to the unobserved region, highlighting the adaptability and efficiency of the latent modulation space. Further, this property is essential for scientific applications where one frequently encounters novel or intermediate parameter regimes that were not explicitly present in the training set. Additional experiments, including settings where partial observations are available for $\beta^*$, are discussed in Appendix F.2.

## 5.3  Bidirectional Inference

A core contribution of the PDEfuncta framework is its ability to perform bidirectional functional inference between two function spaces $\mathcal{A}$ and $\mathcal{U}$, via a joint latent representation. This capability enables unified modeling of forward and inverse mappings between parametric PDE function spaces — a setting frequently encountered in scientific applications such as aerodynamic design. In this section, we evaluate this bidirectional capability in two key aspects: (i) reconstruction fidelity on seen paired instances using diverse modulation strategies and (ii) generalization to unseen PDE instances with complex geometries in neural operator settings.

Table 3: Comparison with existing modulation methods (PSNR↑/MSE↓)

| Modulation | FWI ($\mathcal{A}$) | | FWI ($\mathcal{U}$) | | Navier–Stokes ($\mathcal{A}$) | | Navier–Stokes ($\mathcal{U}$) | |
|---|---|---|---|---|---|---|---|---|
| | PSNR | MSE | PSNR | MSE | PSNR | MSE | PSNR | MSE |
| Shift [17] | 25.879 | 0.0102 | 24.995 | 0.0126 | 28.134 | 0.0155 | 26.306 | 0.0219 |
| Scale [17] | 25.256 | 0.0117 | 29.978 | 0.0039 | 38.784 | 0.0013 | 38.747 | 0.0013 |
| FiLM [22] | 25.292 | 0.0115 | 30.026 | 0.0035 | 42.649 | 0.0005 | 43.105 | 0.0004 |
| GFM | **28.757** | **0.0054** | **32.761** | **0.0021** | **43.166** | **0.0004** | **43.792** | **0.0003** |

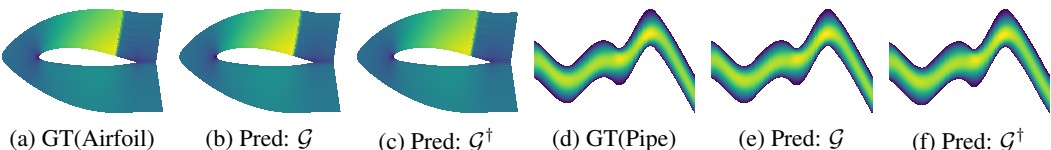

(a) GT(Airfoil)  (b) Pred: $\mathcal{G}$  (c) Pred: $\mathcal{G}^\dagger$  (d) GT(Pipe)  (e) Pred: $\mathcal{G}$  (f) Pred: $\mathcal{G}^\dagger$

Figure 7: PDEfuncta reconstructs both forward $\mathcal{G}^\dagger : \mathcal{A} \to \mathcal{U}$ and inverse $\mathcal{G} : \mathcal{U} \to \mathcal{A}$ mappings between function spaces for unseen airfoil and pipe flow samples.

### 5.3.1 Reconstruction Fidelity on Seen Samples: Modulation Comparison

We evaluate PDEfuncta's ability to compress and reconstruct paired data from seen samples using a shared 20-dimensional latent vector. Experiments are conducted on the FWI and 2D incompressible Navier–Stokes datasets, covering distinct scientific domains. In FWI, $\mathcal{A}$ denotes seismic data and $\mathcal{U}$ is the corresponding velocity map. For Navier–Stokes, $\mathcal{A}$ consists of velocity fields at early time steps ($t \in [0, 14]$), and $\mathcal{U}$ at later timesteps ($t \in [15, 29]$). This setup enables bidirectional inference between physical states over time or domains. We compare four modulation strategies: Shift, Scale, FiLM, and our GFM. As shown in Table 3, GFM achieves the best reconstruction performance across both directions. On FWI, it reconstructs velocity maps from seismic data with a PSNR of 32.76 dB. On Navier–Stokes, it shows the lowest MSE of 0.0003 for velocity reconstruction at $t \in [15, 29]$. These results highlight PDEfuncta's ability to learn accurate bidirectional mappings between paired fields. Additional experimental results and analyses are provided in the Appendix G.

### 5.3.2 Generalization to Unseen Instances: Neural Operator Comparison

Finally, we compare our PDEfuncta with neural operator baselines on two complex geometry benchmark datasets extending it to neural operator, i.e., Euler-NACA (Airfoil) and Pipe. The Euler-NACA dataset predicts airflow ($\mathcal{U}$) over held out airfoil shape ($\mathcal{A}$), whereas the pipe benchmark dataset considers incompressible flow ($\mathcal{U}$) in unseen pipe cross sections($\mathcal{A}$). In both cases, we aim to learn a pair of bidirectional solution operators between function spaces: the forward mapping $\mathcal{G} : \mathcal{A} \to \mathcal{U}$ and the inverse mapping $\mathcal{G}^\dagger : \mathcal{U} \to \mathcal{A}$.

Table 4: Test $L_2$-error ($\downarrow$)

| Method | Euler-NACA | Pipe |
|---|---|---|
| FNO | 0.0385 | 0.0153 |
| UNet | 0.0505 | 0.0298 |
| Geo-FNO | 0.0158 | **0.0066** |
| FFNO | 0.0062 | 0.0073 |
| CORAL | 0.0059 | 0.0120 |
| MARBLE | 0.0058 | 0.0103 |
| PDEfuncta | **0.0051** | 0.0081 |

We evaluate both $\mathcal{G}$ and $\mathcal{G}^\dagger$ on unseen samples. Table 4 reports $L_2$ errors for forward mapping $\mathcal{G}$, and PDEfuncta achieves the lowest error on the Airfoil task and competitive performance on the Pipe dataset. As shown in Figure 7, PDEfuncta demonstrates strong inference performance in both forward and inverse mappings. These results demonstrate that the proposed model generalizes well across domains and supports consistent bidirectional inference between geometry and solution spaces.

## 6 Conclusion

We presented GFM, a simple re-parameterization that handles spectral bias in modulated INRs by conditioning every layer through a fixed Fourier basis. Theory and experiments show that this approach can learn high-frequency components as well as low-frequency components, and provides reliable compression of scientific machine learning data. Building on GFM, we introduced PDEfuncta, a bidirectional novel architecture that encodes paired fields with a single latent vector. Taken together, GFM and PDEfuncta offer a lightweight, resolution free characteristic, unified frameworks for functionalized data and spectrally balanced modeling of PDE systems.

# 7 Acknowledgement

K. Lee acknowledges support from the U.S. National Science Foundation under grant IIS 2338909. This work was also supported by Samsung Research Funding & Incubation Center of Samsung Electronics under Project Number SRFC-IT2402-08, and by Samsung Electronics Co., Ltd. (No. G01240136, KAIST Semiconductor Research Fund (2nd)).

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

# A Symbol Definitions

Table 5 summarizes the key symbols and notation used throughout the paper for clarity and reference.

Table 5: Definitions of symbols.

| Symbol | Description |
|---|---|
| ***Input and function space variables*** | |
| $\mathcal{X}$ | Input coordinate (spatial and/or temporal position). |
| $x$, $t$ | Spatial and temporal components of the input coordinate $X$. |
| $f_a(\mathcal{X}; \theta_a)$ | INR modeling a function in space $\mathcal{A}$ (with parameters $\theta_a$). |
| $f_u(\mathcal{X}; \theta_u)$ | INR modeling a function in space $\mathcal{U}$ (with parameters $\theta_u$). |
| $\mathcal{A}, \mathcal{U}$ | Input and output function spaces |
| $\mathcal{G}, \mathcal{G}^\dagger$ | Function space mapping $\mathcal{A} \to \mathcal{U}$ and $\mathcal{U} \to \mathcal{A}$ |
| ***Neural network architecture parameters*** | |
| $\theta$ | Set of network parameters (weights and biases of an INR). |
| $\theta_a$, $\theta_u$ | Network parameters for $f_a$ and $f_u$, respectively. |
| $L$ | Number of hidden layers in the INR. |
| $h_0$ | Initial input to the network. |
| $h_k$ | Hidden feature vector at layer $k$ ($h_0$ is input, $h_{L+1}$ is final hidden layer output). |
| $W^k$ | Weight matrix of layer $k$ in the network. |
| $b^k$ | Bias vector of layer $k$ in the network. |
| $\gamma$ | Nonlinear activation function applied to hidden layers. |
| $d$ | Dimension of each hidden layer (length of $h_k$ and $b_k$). |
| ***Fourier reparameterization and spectral terms*** | |
| $S^k$ | Trainable coefficient matrix for layer $k$ under Fourier reparameterization. |
| $\Phi_k$ | Fixed Fourier basis for layer $k$. |
| $p_m$ | $m$-th point in the uniform spatial grid used to construct $\Phi_k$. |
| $T_{\max}$ | Maximum coordinate value for the Fourier basis sampling interval. |
| $\omega = \{\omega_{low}, \omega_{high}\}$ | Frequency component used in constructing the Fourier basis. |
| $q$ | Phase shift applied in the Fourier basis construction. |
| $D$ | Number of Fourier basis functions per layer. |
| $M$ | Number of sample points per basis function. |
| $F_1$, $F_2$ | High and low reference frequencies ($F_1 > F_2 > 0$) used in Theorem 1. |
| $\epsilon$ | Small positive constant (error tolerance in Theorem 1). |
| ***Modulation and training variables*** | |
| $R^k$ | Shared base coefficient matrix for layer $k$. |
| $\alpha^k$ | Modulation vector for layer $k$ |
| $\alpha^{k,W}$, $\alpha^{k,b}$ | Components of $\alpha_k$ that modulate the weight and bias at layer $k$, respectively. |
| $\psi_k(\cdot)$ | Modulation function applied at layer $k$ |
| $z$ | Latent modulation vector (sample-specific) |
| $g_a(z; \pi_a)$ | Mapping for function space $\mathcal{A}$: generates $\{\alpha_k^a\}_{k=1}^L$ from latent $z$. |
| $g_u(z; \pi_u)$ | Mapping for function space $\mathcal{U}$: generates $\{\alpha_k^u\}_{k=1}^L$ from latent $z$. |
| $\pi_a$, $\pi_u$ | Learnable parameters of the modulation networks $g_a$ and $g_u$. |
| $\eta_{\text{inner}}, \eta_{\text{outer}}$ | Learning rates for inner/outer loop |
| $\mathbb{L}$ | Training loss function. |
| ***PDE-specific parameters (experiment settings)*** | |
| $\beta$ | PDE coefficient in the 1D convection equation. |
| $a_1$, $a_2$ | PDE coefficients in the 2D Helmholtz equations. |

## B  Limitation and Broader Impact

Our proposed GFM method allows explicit control over the ratio of high-frequency and low-frequency components in the Fourier basis during weight construction. While optimal performance may require careful hyperparameter tuning to select the appropriate frequency ratio, this flexibility enables our framework to be adapted to a wide range of scientific problems and data characteristics. Additionally, our proposed methods, GFM and PDEfuncta, enable efficient compression and representation of multiple PDE solution datasets, offering significant advantages in terms of memory efficiency. These methods can facilitate effective storage and transmission of scientific data, potentially making large-scale modeling and simulation more accessible across various domains.

## C  Comparison with Existing Neural PDE Solvers

PDEfuncta demonstrates clear advantages over traditional Physics-Informed Neural Networks (PINNs) and Neural Operators. (1) PINNs, while effective in integrating physical laws directly into training, often struggle with scalability and generalization to complex, high-frequency PDE solutions. (2) Neural Operators, including methods like FNO and DeepONet, address scalability but typically require consistent discretizations and are limited to one-way function space mappings. PDEfuncta overcomes these challenges by leveraging Global Fourier Modulation, providing a compact, efficient, and spectrally-aware representation capable of accurate forward and inverse mappings without retraining. Its shared latent modulation vector further enhances generalization across varied parameter spaces, demonstrating clear advantages in efficiency, generalization, and bidirectional inference capabilities compared to both PINN and Neural Operator frameworks.

## D  Dataset Description

We provide detailed descriptions of all benchmark PDE datasets used in the main paper, including the governing equations, boundary/initial conditions, parameter ranges, and scientific context for each.

### D.1  Convection Equation

The convection dataset is based on the one-dimensional convection equation with a controllable advection speed parameter $\beta$. The PDE governs the evolution of a field $u(x, t)$ as:

$$\frac{\partial u}{\partial t} + \beta \frac{\partial u}{\partial x} = 0, \quad x \in \Omega, \ t \in [0, T], \tag{13}$$

where $\beta$ is the convection coefficient. In our experiments, $\beta$ varies within a range (e.g. $\beta \in [1, 50]$) to represent solution fields of different characteristic speeds. We impose periodic boundary conditions on the spatial domain $\Omega$. An initial condition $u(x, 0)$ is $1 + \sin(x)$). It is a fundamental model for wave propagation and convective transport.

### D.2  Helmholtz Equation

The Helmholtz dataset involves a steady-state wave equation in two spatial dimensions. We generate solutions of the form $u(x, y)$ by choosing forcing functions that yield analytic solutions. The PDE is given by:

$$\frac{\partial^2 u(x, y)}{\partial x^2} + \frac{\partial^2 u(x, y)}{\partial y^2} + k^2 u(x, y) - q(x, y) = 0,$$
$$q(x, y) = (-(a_1\pi)^2 - (a_2\pi)^2 + k^2)\sin(a_1\pi x)\sin(a_2\pi y), \tag{14}$$

$$u(x, y) = k^2 \sin(a_1\pi x)\sin(a_2\pi y), \tag{15}$$

where $k$ is the wavenumber and $q(x, y)$ is a source term. The coefficients $a_1$ and $a_2$ are PDE coefficients that control the number of oscillations of the solution in the $x$ and $y$ directions, respectively. In our dataset, we consider two difficulty settings: Helmholtz #1 with $\{a_1, a_2\} \in \{1, \ldots, 5\}$ (limited frequencies), and Helmholtz #2 with $\{a_1, a_2\} \in \{1, \ldots, 10\}$.

### D.3 Kuramoto-Sivashinsky Equation

The one-dimensional Kuramoto–Sivashinsky equation [43] is a prototypical model for spatio-temporal chaos and pattern formation. Following the experimental setup in [43], we use the data without normalization to preserve its natural scale.

$$u_t + uu_x + u_{xx} + \nu u_{xxxx} = 0, \quad (t, x) \in [0, T] \times [0, L],$$

$$u(0, x) = u_0(x), \quad u_0(x) = \sum_{i=0}^{20} A_i \sin((2\pi k_i x / L) + \phi_i), \tag{16}$$

where $u(x, t)$ is the evolving scalar field, and $\nu > 0$ is a viscosity (or hyperviscosity) parameter. In our simulations, we take periodic boundary conditions on the domain $[0, L]$ (so that derivatives in $x$ are periodic), which is natural given that initial conditions are composed of Fourier modes.

### D.4 Navier-Stokes Equation

We use the 2D Navier–stokes equation data employed in the neural operator related studies [20, 21, 44]. This equation represents the dynamics of incompressible fluid within a rectangular domain and is expressed as follows:

$$\frac{\partial w(x, t)}{\partial t} = -u(x, t) \cdot \nabla w(x, t) + \nu \Delta w(x, t) + f, \qquad x \in (0, 1)^2, \, t \in (0, T] \tag{17}$$

$$w(x, t) = \nabla \times u(x, t), \qquad x \in (0, 1)^2, \, t \in (0, T] \tag{18}$$

$$\nabla \cdot u(x, t) = 0, \qquad x \in (0, 1)^2, \, t \in (0, T] \tag{19}$$

where $u$ is the velocity field and $w$ is the vorticity. $\nu$ is the viscosity and $f$ is a forcing term. The initial condition $\omega_0(x)$ is generated according to $\omega_0 \sim \mu$ where $\mu = \mathcal{N}(0, 7^{3/2}(-\Delta + 49I)^{-2.5})$.

We consider the spatial domain $x \in (0, 1)^2$ with the following boundary condition:

$$f(x_1, x_2) = 0.1(\sin(2\pi(x_1 + x_2)) + \cos(2\pi(x_1 + x_2))). \tag{20}$$

### D.5 Euler's Equation (Airfoil)

In our study, we employ the Euler's equation (airflow), which is a pivotal component of the discussed benchmark data in [34, 22]. The governing equation is as follows:

$$\frac{\partial \rho_f}{\partial t} + \nabla \cdot (\rho_f \mathbf{v}) = 0, \quad \frac{\partial \rho_f \mathbf{v}}{\partial t} + \nabla(\cdot \rho_f \mathbf{v} \otimes \mathbf{v} + p\mathbb{I}) = 0, \quad \frac{\partial E}{\partial t} + \nabla \cdot ((E + p)\mathbf{v}) = 0, \tag{21}$$

where $\rho_f$, $p$, $\mathbf{v}$ and $E$ are the fluid density, the pressure, the velocity vector and the total energy respectively. The viscous effect is neglected. The far-field boundary condition is specified as $\rho_\infty = 1$, $p_\infty = 1.0$, $M_\infty = 0.8$, $AoA = 0$, where $M_\infty$ represents the Mach number and $AoA$ denotes the angle of attack. In our experimentation, we use a total of 1,000 training data samples and 200 test data samples. The computational grid utilized is the C-grid mesh with about $200 \times 50$ quadrilateral elements.

### D.6 Pipe

The pipe dataset [35] is derived from simulations of viscous incompressible flow in a pipe, governed by the Navier–Stokes equations for velocity and pressure.

$$\frac{\partial \mathbf{v}}{\partial t} + (\mathbf{v} \cdot \nabla)\mathbf{v} = -\nabla p + \nu \nabla^2 \mathbf{v}, \tag{22}$$

$$\nabla \cdot \mathbf{v} = 0$$

where $\mathbf{v}$ is the velocity field, $p$ is the pressure, and $\nu$ is the kinematic viscosity. The domain is a cylindrical pipe, a long tube with circular cross section, though our dataset is represented on a mesh (in the format of a $129 \times 129$ grid for a cross-sectional plane, as indicated by the input size).

### D.7 Seismic Wave Equation

The OpenFWI dataset [10] is a collection of seismic forward modeling simulations used for full-waveform inversion research. Each data sample consists of a subsurface velocity model (a spatial map of wave propagation speeds) and simulated seismic wavefields recorded from sources propagating through that medium.

$$\nabla^2 p - \frac{1}{v^2}\frac{\partial^2 p}{\partial t^2} = s, \tag{23}$$

where $p(x, z, t)$ is the pressure wavefield (or particle displacement) at spatial location $(x, z)$ and time $t$, and $v(x, z)$ is the velocity model (spatially-varying wave speed). The term $s(x, z, t)$ represents the seismic source (for example, a Ricker wavelet point source on or near the surface). In our dataset, $\Omega$ is a 2D slice of the Earth's subsurface (with $x$ horizontal and $z$ depth).

## E  Detailed experimental setup

This section provides the detailed experimental setup, including hyperparameter choices and computational cost measurements. Experiments are conducted on a system running UBUNTU 18.04 LTS, PYTHON 3.9.7, PYTORCH 1.13.0, CUDA 11.6, i9 CPU, and NVIDIA RTX A5000.

### E.1  Hyperparameters

For all experiments, we fixed the backbone INR to SIREN in order to isolate and compare the effects of different modulation methods. Across all models and experiments, we set $\eta_{inner} = 0.01$ and $\eta_{outer}$ = 0.0001. All networks use a total of 5 layers, each with hidden dimension $M = 256$. The modulation mapping is implemented as a two-layer MLP with hidden dimension 512, and the dimension of the latent code $z$ is fixed to 20 for all settings. For GFM, additional hyperparameter settings are required to construct the Fourier basis (cf. Appendix H). Specifically, the number of phase shifts ($n_{\text{phase}}$) is fixed to 32 for all datasets, while the numbers of high-frequency and low-frequency components ($n_{\text{high}}$, $n_{\text{low}}$) are set as follows: 128 and 32 for Convection and FWI, 128 and 128 for Helmholtz, 64 and 16 for Kuramoto-Sivashinsky, and 8 and 128 for Navier-Stokes. The batch size and the number of training epochs for each dataset are as follows: batch size 32 and epoch 1,000 for Convection, batch size 16 and epoch 5,000 for Helmholtz, bach size 32 and epoch 5,000 for Kuramoto-Sivashinsky, batch size 10 and epoch 1,000 for Navier-Stokes, and batch size 16 and epoch 10,000 for FWI.

### E.2  Computational cost

To assess the computational requirements of different modulation methods, we report peak GPU memory usage and average training time per epoch for each model, measured on a single NVIDIA A5000 GPU. As a representative case, we provide results for the Helmholtz#2 setting under the single-INR modulation configuration (cf. Section 5.2 and Appendix J). Results are summarized in Table 6. GFM achieves a strong trade-off between model complexity and com-

Table 6: Computational cost

| Method | Memory (GB) | Time (sec) |
|---|---|---|
| Shift | 8.15 | 0.20 |
| Scale | 12.15 | 0.24 |
| FiLM | 12.15 | 0.29 |
| SpatialFuncta | 8.83 | 0.59 |
| GFM (Ours) | 8.28 | 0.27 |

putational efficiency, with memory consumption and training speed close to Shift, yet substantially better than Scale, FiLM, and SpatialFuncta. Notably, SpatialFuncta exhibits a much higher training time per epoch (0.59s), which is due to the extra computational cost of applying modulation at every spatial location, rather than globally, despite moderate memory usage. Other experimental settings showed similar trends in both memory and runtime.

# F   More Experimental Results on Experiments with Single-INR modulation

This section provides additional analyses and experimental results to supplement Section 5.2. We present training loss curves for every dataset, a detailed evaluation of the latent modulation space through interpolation analyses, and statistical comparisons across all benchmarks.

## F.1   Training loss curves

Figure 8 presents the training loss (MSE) curves for Shift, Scale, FiLM, and GFM across all PDE benchmarks considered in Section 5.2. As discussed in Section 5.2, GFM consistently achieves faster and more stable convergence compared to other modulation strategies. This advantage is especially notable for challenging cases such as Helmholtz#2 and Kuramoto-Sivashinsky, where baselines exhibit slow or unstable training. These results further demonstrate the effectiveness of GFM in robust optimization and in mitigating the spectral bias typically observed in modulated INRs.

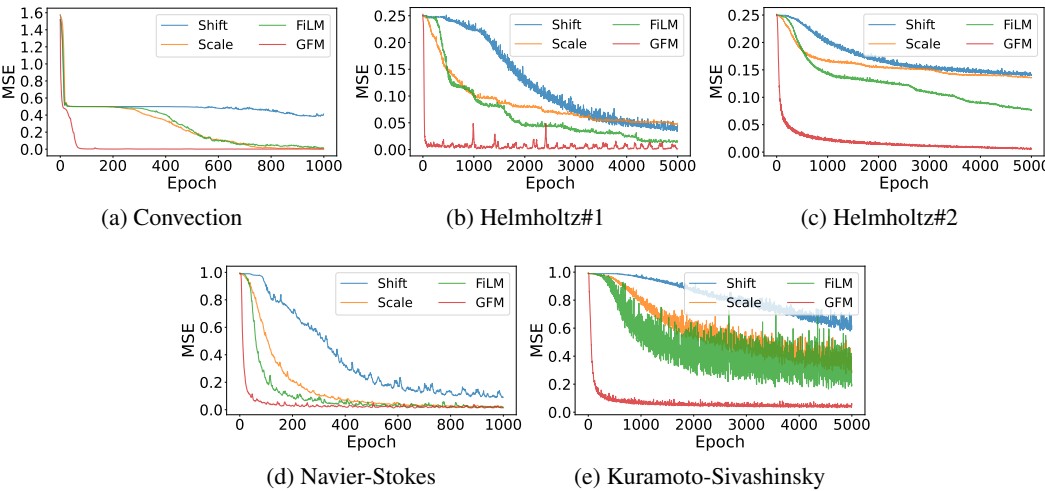

Figure 8: Training loss (MSE) curves for Shift, Scale, FiLM, and GFM on (a) Convection, (b) Helmholtz#1, (c) Helmholtz#2, (d) Navier-Stokes, and (e) Kuramoto-Sivashinsky. GFM consistently achieves faster and more stable convergence across all benchmarks.

## F.2   Additional Analysis of Latent Modulation Space

To further analyze the properties of the learned latent modulation space, we present additional qualitative and quantitative results for the generalization experiments described in Section 5.2.1. Specifically, we visualize the dynamics of the interpolated latents for the convection equation and compare interpolation results across different modulation strategies.

Figure 9 shows the cubic interpolation trajectories of the latent code $z$ for the convection equation. Each subplot corresponds to one dimension of the 20-dimensional latent modulation vector. Blue dots indicate latents obtained from the seen (training) coefficients, while red crosses denote interpolated latents for unseen test coefficients, estimated via cubic interpolation. This visualization suggests that the latent space encodes smooth and coherent functional variations as the PDE parameter is varied. The interpolated latents for unseen coefficients generally follow the manifold traced by the seen points, indicating that the latent space may provide a functionally meaningful representation aligned with the underlying parametric dependence of the PDE family.

To further investigate the effect of latent interpolation across different modulation strategies, Figure 10 presents the reconstruction results for an unseen coefficient ($\beta = 30.5$) obtained via cubic interpolation of the latent code for each modulation method (Shift, Scale, FiLM, and GFM). The ground truth solution is shown in Figure 10(a), and Figures 10(b)–(e) show the corresponding reconstructions. These results highlight that only GFM produces an accurate reconstruction that closely matches the ground truth, whereas Shift, Scale, and FiLM all fail to capture the solution structure.

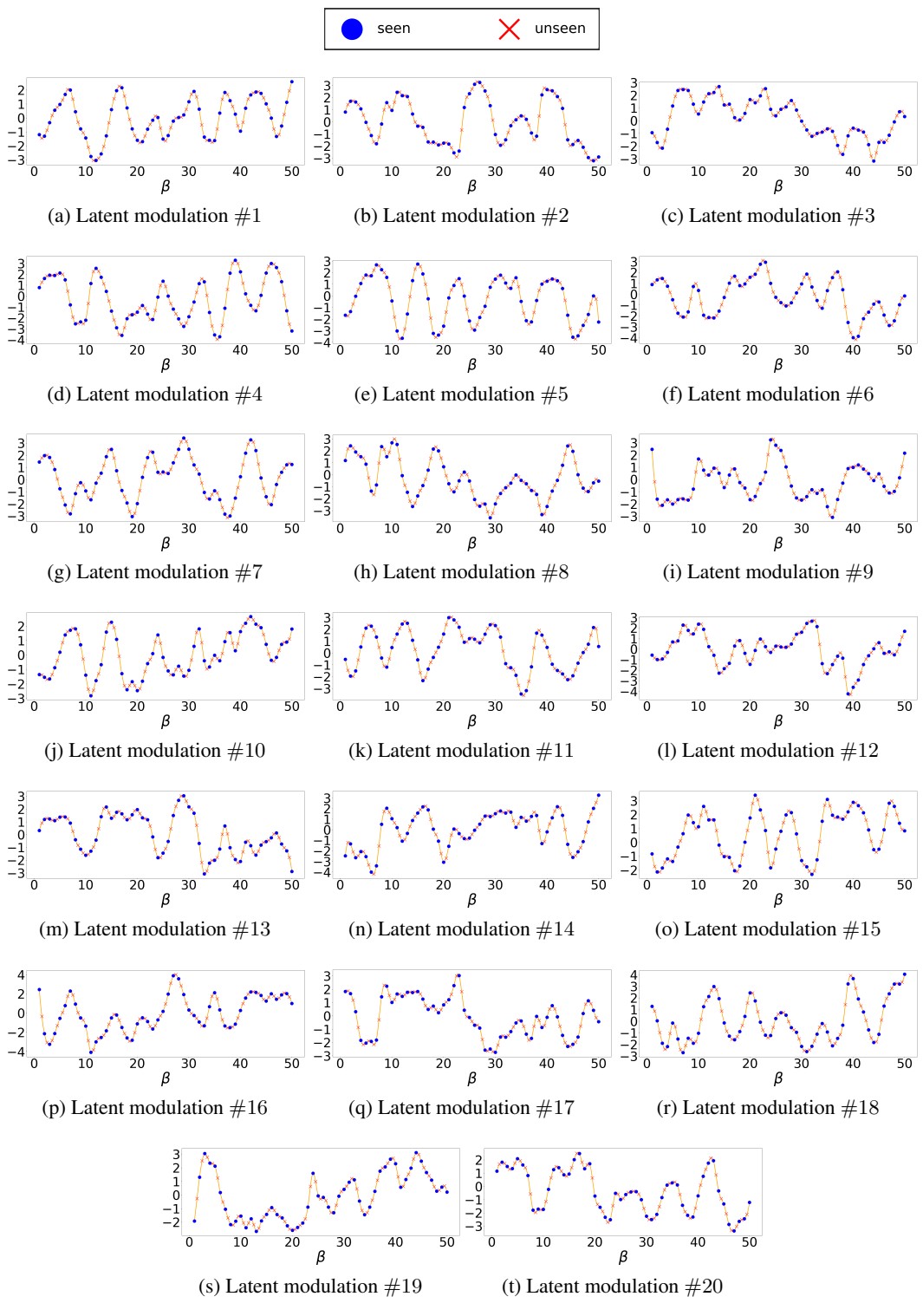

Figure 9: Dynamics of Latents (convection equations)

## F.3 Statistical Comparison Across Modulation Methods

Table 7 provides the mean and standard deviation of PSNR across three trials for each modulation method and PDE benchmark. While Table 1 in the main paper presents the mean values, this supplementary table highlights the consistency and robustness of the results by reporting the standard deviations.

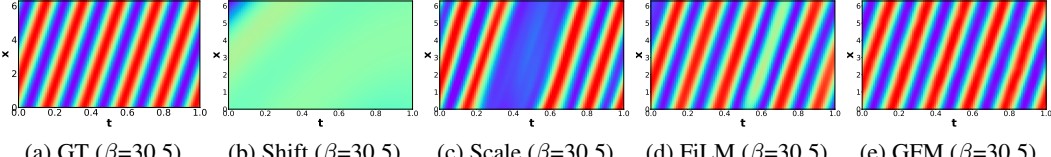

| (a) GT ($\beta$=30.5) | (b) Shift ($\beta$=30.5) | (c) Scale ($\beta$=30.5) | (d) FiLM ($\beta$=30.5) | (e) GFM ($\beta$=30.5) |

Figure 10: **[Convection] Predicted solution snapshots for unseen coefficient** $\beta = 30.5$ **from interpolated latent codes using different modulation methods.** (a) Ground truth, (b) Shift, (c) Scale, (d) FiLM, and (e) GFM. For each method, the latent code for $\beta = 30.5$ is obtained via cubic interpolation between neighboring training latents.

Table 7: Comparison with existing modulation methods with mean and standard deviation (PSNR↑).

| Modulation | Convection | Helmholtz #1 | Helmholtz #2 | Navier-Stokes | Kuramoto-Sivashinsky |
|---|---|---|---|---|---|
| Shift [17] | 13.225±0.597 | 25.467±0.925 | 18.249±0.901 | 25.767±0.635 | 16.962±0.819 |
| Scale [17] | 27.410±0.608 | 25.669±0.492 | 18.183±0.683 | 31.469±0.507 | 20.395±0.997 |
| FiLM [22] | 27.330±1.254 | 33.611±0.506 | 22.771±0.533 | 31.904±0.502 | 22.297±1.896 |
| GFM | 42.474±0.412 | 39.139±0.530 | 29.033±0.556 | 34.754±0.779 | 29.827±0.290 |

# G   More Experimental Results on Bidirectional Inference

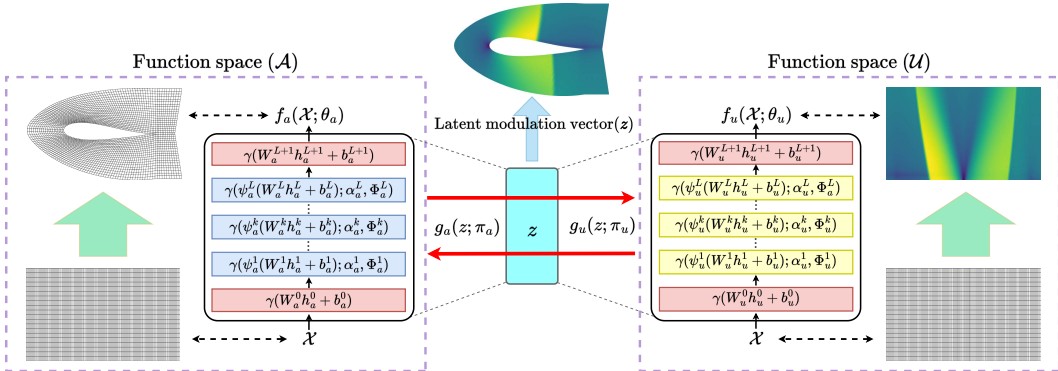

Figure 11: **Visualization of the PDEfuncta pipeline for the Airfoil dataset.** Two modalities in airfoil data are represented as separate INRs $f_a$ and $f_u$, and are jointly compressed and reconstructed through a shared latent modulation vector $z$.

This section provides supplementary explanations for Section 5.3. First, we include a schematic illustration to visually describe how the Airfoil dataset is used in the unseen neural operator setting (cf. Section 5.3.2). Figure 11 illustrates the overall architecture of PDEfuncta applied to the Airfoil dataset. The two modalities–geometry and flow field–are modeled using separate INRs $f_a$ and $f_u$, while a shared latent modulation vector enables joint compression and bidirectional reconstruction between them. Additionally, we present FWI reconstruction results for seen samples using different modulation methods in Figures 12 and 13.

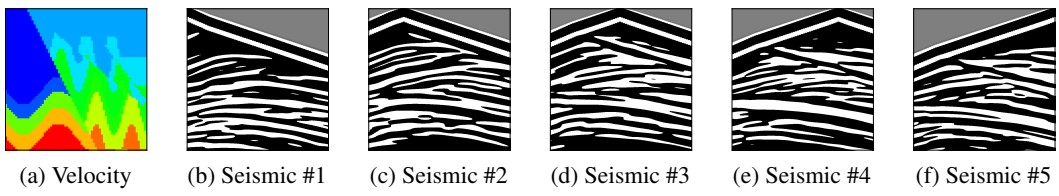

(a) Velocity    (b) Seismic #1    (c) Seismic #2    (d) Seismic #3    (e) Seismic #4    (f) Seismic #5

Figure 12: **Ground truth for a representative FWI sample.** (a) Velocity map and (b–f) corresponding seismic data.

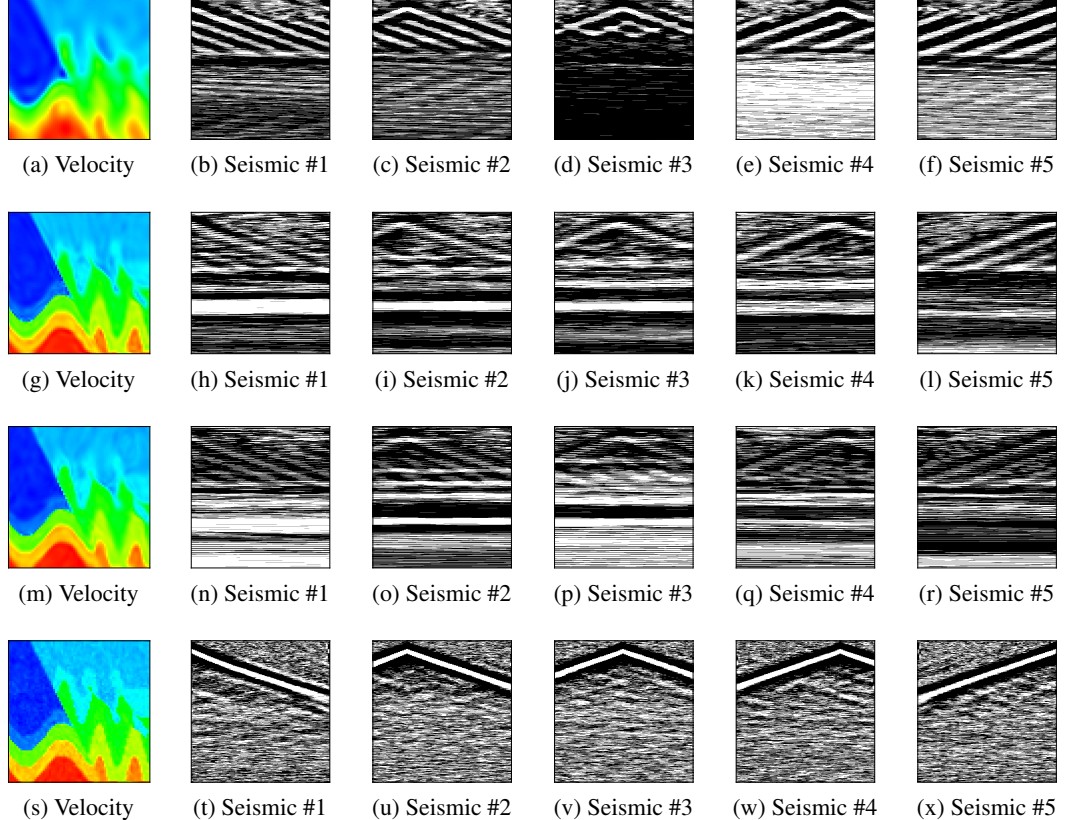

(a) Velocity    (b) Seismic #1    (c) Seismic #2    (d) Seismic #3    (e) Seismic #4    (f) Seismic #5

(g) Velocity    (h) Seismic #1    (i) Seismic #2    (j) Seismic #3    (k) Seismic #4    (l) Seismic #5

(m) Velocity    (n) Seismic #1    (o) Seismic #2    (p) Seismic #3    (q) Seismic #4    (r) Seismic #5

(s) Velocity    (t) Seismic #1    (u) Seismic #2    (v) Seismic #3    (w) Seismic #4    (x) Seismic #5

Figure 13: **FWI bidirectional reconstruction results for seen samples using different modulation methods.** Each row shows a velocity map and its corresponding seismic data reconstructed by a different modulation method: (a–f) Shift, (g–l) Scale, (m–r) FiLM, and (s–x) GFM.

# H Implementation Details of Fourier Basis Construction

This section provides additional details regarding the construction and implementation of the fixed Fourier bases used in Global Fourier Modulation (GFM), supplementing Section 3.1 of the main paper. The fixed Fourier bases for each layer are constructed following the procedure introduced in [38]. In all experiments, the fixed Fourier basis $\Phi^k$ for each layer is constructed by combining a set of frequencies and phase shifts (cf. Equation 3.1). To ensure spectral diversity, we employ $n_{\text{low}}$ low-frequency components ($\omega_{\text{low}} = \{\frac{1}{n_{\text{low}}}, \frac{2}{n_{\text{low}}}, ..., 1\}$) and $n_{\text{high}}$ high-frequency components ($\omega_{\text{high}} = \{1, 2, ..., n_{\text{high}}\}$). For each frequency, we use $n_{\text{phase}}$ phase shifts uniformly spaced over $[0, 2\pi]$. Here, $n_{\text{low}}$, $n_{\text{high}}$, and $n_{\text{phase}}$ are hyperparameters, and the total number of basis vectors per layer is thus $D = (n_{\text{low}} + n_{\text{high}}) \times n_{\text{phase}}$. Each basis vector is evaluated at uniformly spaced $M$ points within the interval $[-T_{\max}/2, T_{\max}/2]$, where $T_{\max} = 2\pi n_{\text{low}}$. The number of sampling points $M$ is chosen to match the input dimension of the corresponding layer, ensuring compatibility with weight reparameterization. In our implementation, these bases are precomputed, fixed throughout training, and not updated by gradient descent.

# I Meta-learning based Training Algorithm

In this section, we outline the meta-learning-based training and inference procedures for PDEfuncta. Algorithm 1 describes both the training and inference phase, where both network parameters and sample-specific latent vectors are jointly optimized via a nested inner–outer loop.

---

**Algorithm 1** Training and Inference of the proposed method

---

1: /* Training */
2: **Input:** Spatio-temporal grid $\mathcal{X} = (x, t)$
3: Randomly initialize $\theta = \{\theta_a, \theta_u\}, \pi = \{\pi_a, \pi_u\}$ and set $z_{train} \leftarrow 0$ (zero vector)
4: **while** not done **do**
5:     Sample batch $\mathcal{B}$ of output $\{a^i, u^i\}_{i \in \mathcal{B}}$
6:     Sample $K$ examples from batch $\mathcal{B}$
7:     /* Inner loop */
8:     **for** $j = 1$ to $K$ **do**
9:         $\mathbb{L}_j = \mathbb{L}_{\mathcal{X}_j}(a^j, f_a(\mathcal{X}; \theta_a, g_a(z_{train}^j; \pi_a))) + \mathbb{L}_{\mathcal{X}_j}(u^j, f_u(\mathcal{X}; \theta_u, g_u(z_{train}^j; \pi_u)))$
10:        $z_{train}^j \leftarrow z_{train}^j - \eta_{inner} \nabla_{z_{train}^j} \mathbb{L}_j$
11:     **end for**
12:     /* Outer loop */
13:     $\mathbb{L}_{\mathcal{B}} = \sum_{i \in \mathcal{B}}(\mathbb{L}_{\mathcal{X}_i}(a^i, f_a(\mathcal{X}; \theta_a, g_a(z_{train}^i; \pi_a))) + \mathbb{L}_{\mathcal{X}_i}(u^i, f_u(\mathcal{X}; \theta_u, g_u(z_{train}^i; \pi_u))))$
14:     $\theta \leftarrow \theta - \eta_{outer} \nabla_\theta \mathbb{L}_{\mathcal{B}}$
15:     $\pi \leftarrow \pi - \eta_{outer} \nabla_\pi \mathbb{L}_{\mathcal{B}}$
16: **end while**
17:
18: /* Inference ($\mathcal{G} : \mathcal{A} \to \mathcal{U}$) */
19: **Input:** New sample $a^{test}$ from space $\mathcal{A}$, coordinates $\mathcal{X}$, trained parameters $\theta, \pi$
20: **Output:** Corresponding sample $u^{pred}$ in space $\mathcal{U}$
21: /* *Latent Fitting* */
22: **while** not converged **do**
23:     $\mathbb{L}_a = \mathbb{L}_{\mathcal{X}}(a^{test}, f_a(\mathcal{X}; \theta_a, g_a(z_{test}; \pi_a)))$
24:     $z_{test} \leftarrow z_{test} - \eta_{infer} \nabla_{z_{test}} \mathbb{L}_a$
25: **end while**
26: /* *Prediction* */
27: $u^{pred} = f_u(\mathcal{X}; \theta_u, g_u(z_{test}; \pi_u))$
28: **return** $u^{pred}$
29: /* *Note: Inverse Problem ($\mathcal{G}^\dagger : \mathcal{U} \to \mathcal{A}$) is symmetric: optimize $z_{test}$ using $\mathbb{L}_u = \mathbb{L}_{\mathcal{X}}(u^{test}, f_u(\mathcal{X}; \theta_u, g_u(z_{test}; \pi_u)))$, then predict $a^{pred} = f_a(\mathcal{X}; \theta_a, g_a(z_{test}; \pi_a))$* */

---

## J Comparison with Spatial Functa

Table 8: Performance comparison (PSNR↑/MSE↓) between Spatial Functa and GFM on Convection, Helmholtz, and Kuramoto-Sivashinsky equations.

| Modulation | Convection | | Helmholtz #1 | | Helmholtz #2 | | Kuramoto-Sivashinsky | |
|---|---|---|---|---|---|---|---|---|
| | PSNR | MSE | PSNR | MSE | PSNR | MSE | PSNR | MSE |
| SpatialFuncta [37] | 20.390 | 0.0739 | 28.908 | 0.0068 | 25.976 | 0.0141 | 20.728 | 3.4297 |
| GFM | **42.474** | **0.0004** | **39.139** | **0.0006** | **29.033** | **0.0056** | **29.827** | **0.4198** |

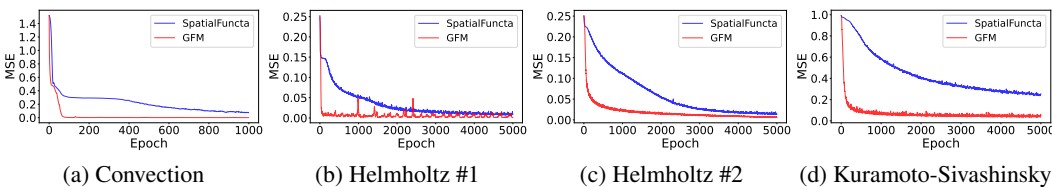

(a) Convection     (b) Helmholtz #1     (c) Helmholtz #2     (d) Kuramoto-Sivashinsky

Figure 14: Train loss (MSE) curves comparing Spatial Functa and GFM across epochs for PDE benchmarks (Convection, Helmholtz#1, Helmholtz#2, Kuramoto-Sivachinsky)

To evaluate the performance of our proposed GFM in comparison with Spatial Functa [37], we conduct experiments under the same conditions as those reported in Table 1. Unlike global modulation approaches such as Shift, Scale, FiLM and GFM, which inject instance-specific information via a low-dimensional global latent vector (e.g., $z \in \mathbb{R}^{20}$ in our experimental setup), Spatial Functa utilizes a high-dimensional spatially structured latent code (e.g., $z \in \mathbb{R}^{8 \times 8 \times 16}$ in our implementation) designed to modulate local regions of the underlying function space. Note that these latent dimensions are representative of the settings used in our experiments and may vary depending on the specific model architecture or dataset. Due to the structural difference in latent space dimensionality, a direct comparison under fixed latent size is infeasible. We therefore present Spatial Functa as a separate baseline, following the original implementation and latent dimension, and report results on the convection, Helmholtz, and Kuramoto–Sivashinsky benchmarks. We exclude Navier–Stokes since its 3D-coordinates are incompatible with the 2D latent grid arrangement used by Spatial Functa.

Table 8 reports the reconstruction accuracy for both methods, and Figure 14 shows the training loss curves. Figures 15 and 16 provide the reconstructed solutions on convection and Helmholtz dataset, respectively. Notably, in Figures 15 and 16, we observe that as $\beta$ and $a_1$ increases, the reconstruction quality of Spatial Functa (second row) deteriorates significantly, whereas GFM (third row) maintains high fidelity even in challenging high-frequency regimes. While Spatial Functa achieves moderate improvements over basic global modulation methods such as Shift, Scale, and FiLM (cf. Table 1), our GFM method consistently achieves superior PSNR and lower reconstruction errors across all considered datasets. This empirical advantage highlights the effectiveness of explicitly injecting frequency-aware priors into the modulation space, allowing GFM to better capture both global structure and fine-scale, which global modulation was previously thought to be insufficient.

We also observe that Spatial Functa requires significantly higher computational cost compared to global modulation methods. The use of high-dimensional spatial latent codes results in longer training times (see Section E.2). In contrast, GFM requires less computational cost, leveraging compact latent codes for faster training and inference.

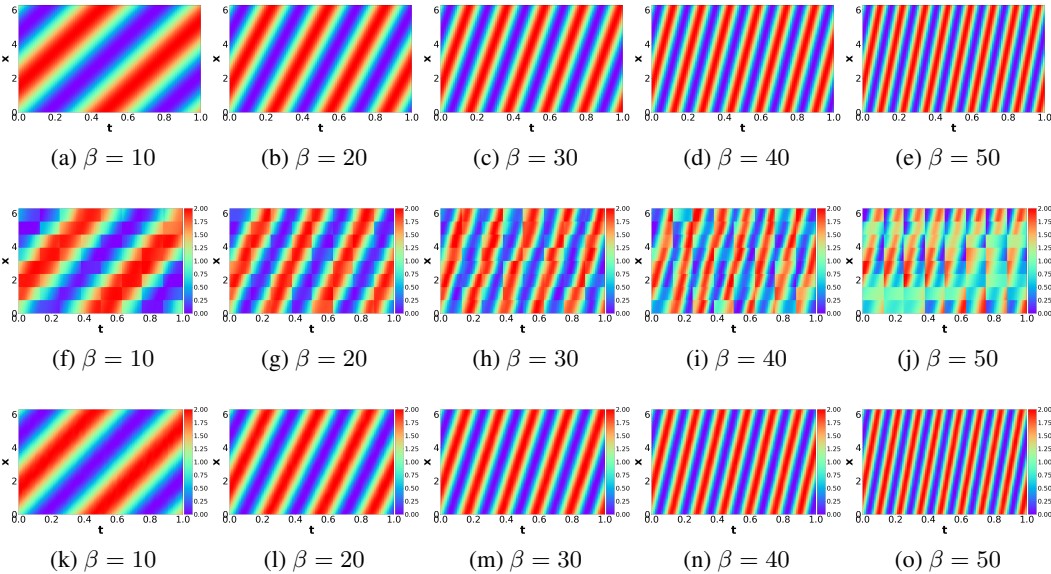

Figure 15: Reconstruction results for Spatial Functa and GFM on the convection equation with $\beta = \{10, 20, 30, 40, 50\}$. Rows represent ground truth (top), Spatial Functa (middle), and GFM (bottom).

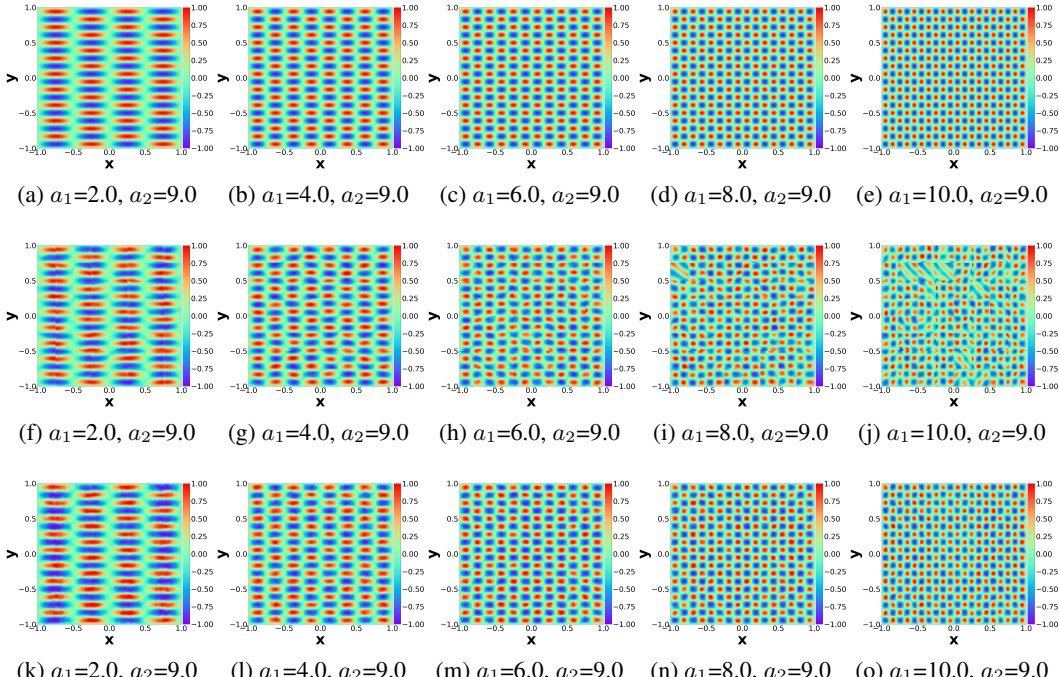

Figure 16: Reconstruction results for Spatial Functa and GFM on the convection equation with $(a_1, a_2) = \{(2.0, 9.0), (4.0, 9.0), (6.0, 9.0), (8.0, 9.0), (10.0, 9.0)\}$. Rows represent ground truth (top), Spatial Functa (middle), and GFM (bottom).

