# OpenReview forum: "PDEfuncta: Spectrally-Aware Neural Representation for PDE Solution Modeling"
_NeurIPS.cc/2025/Conference — NeurIPS 2025 poster_

### Official Review · Reviewer_tDdd · 2025-06-29

**Clarity:** 3
**Significance:** 2
**Originality:** 2
**Rating:** 4
**Confidence:** 5

**Summary:**

This paper deals with the high frequency feature of solutions of some PDEs, e.g., Helmholtz equations. The proposed approach is based on two components: 1. The Functa framework [1] of modulating implicit neural representations (INR); 2. Fourier reparameterization [2]. The latter technique is known for capturing high frequency features with theoretic guarantee.

In experiments, the authors designed two sets of study: 1. Compression and reconstruction; 2. Bidirectional inference. The proposed method outperforms baselines in both ways.

[1] Emilien Dupont, Hyunjik Kim, SM Eslami, Danilo Rezende, and Dan Rosenbaum. From
data to functa: Your data point is a function and you can treat it like one. arXiv preprint
arXiv:2201.12204, 2022.

[2] Kexuan Shi, Xingyu Zhou, and Shuhang Gu. Improved implicit neural representation with
fourier reparameterized training. In Proceedings of the IEEE/CVF Conference on Computer
Vision and Pattern Recognition, pages 25985–25994, 2024.

**Questions:**

N.A.

**Ethical Concerns:**

["NO or VERY MINOR ethics concerns only"]

**Final Justification:**

After checking other reviews, I didn't find any unresolved major concern. However, my opinion about the limitation on scale also remains, as the author admitted MLP being the main type of architecture. Therefore, my score remains to 4.

**Limitations:**

N.A.

**Paper Formatting Concerns:**

N.A.

**Quality:**

3

**Strengths And Weaknesses:**

**Strength**

Quality: The paper demonstrates GFM's superior accuracy, improving PSNR and reducing MSE on 5 types of problems, e.g., convection, Helmoltz1&2, Navier-Stokes, KS (Table 1), and shows competitive performance against neural operator baselines on Euler-NACA and pipeline datasets (Table 4).

Clarity: The paper is logically organized with clear problem statements and effectively uses figures (e.g., Figure 2 for GFM pipeline) to illustrate concepts and results.

Significance: This work features bidirectional inference. It tackles the significant problem of spectral bias in INRs and introduces the PDEfuncta framework, enabling novel multi-modal and bidirectional functional inference between paired function spaces.

Originality: The Global Fourier Modulation (GFM) is a novel contribution extending Fourier reparameterization to modulated INRs, and its integration into the PDEfuncta framework for bidirectional inference is a novel architectural design.

**Weakness**

1. The main concern is about scalability. The proposed method belongs to INR category which is good at capturing details and high frequency feature. However, the architecture of INR mostly relies on MLP, and in the scenario of PDE, how scalable is such method? This paper does not include related analysis.
2. In section 4.2, the optimization variables of inner loop and outer loop are not mentioned, causing a little confusion. The variable of inner loop should be $z$, and other variables will be optimized in outer loop according to Algo. 1 in Appendix. A simple explanation in Section 4.2 could be more convenient to readers.

---

> ### Author Rebuttal · Authors · 2025-07-31
>
> **Q1) The main concern is about scalability. The proposed method belongs to INR category which is good at capturing details and high frequency feature. However, the architecture of INR mostly relies on MLP, and in the scenario of PDE, how scalable is such method? This paper does not include related analysis.**
>
> Currently, coordinate-based neural networks (e.g., INRs, PINNs, NeRFs) predominantly use MLPs as their backbone, since alternative architectures such as CNNs, RNNs, or Transformers typically assume specific structural priors in the data that are not present in coordinate-based representations. Recent advances in network design have introduced the Kolmogorov–Arnold Network (KAN) [1], inspired by the Kolmogorov–Arnold representation theorem. KANs have been proposed as lightweight and potentially more expressive alternatives to MLPs. However, their use in this context is still at an early stage and requires further investigation. Given this current landscape, we adopt MLPs in our work and explore the mentioned direction through additional numerical experiments. The results of the additional experimentation are as follows.
>
> [1] Liu, Ziming, et al. "Kan: Kolmogorov-arnold networks." arXiv preprint arXiv:2404.19756 (2024).
>
> ---
>
> **Q2) In section 4.2, the optimization variables of inner loop and outer loop are not mentioned, causing a little confusion. The variable of inner loop should be , and other variables will be optimized in outer loop according to Algo. 1 in Appendix. A simple explanation in Section 4.2 could be more convenient to readers.**
>
> We will enhance the explanation of the meta-learning algorithm in Section 4.2 by clearly specifying the optimization variables used in the inner and outer loops. Specifically, we will state that the inner loop optimizes only the latent modulation vector z individually for each data sample, while the outer loop optimizes the shared parameters of the INR networks $\theta_a$, $\theta_u$, and the modulation network parameters $\pi_a$, $\pi_u$. This clarification will align the main text with Algorithm 1 in the Appendix I, improving readability and comprehension.

---

> > ### Comment · Reviewer_tDdd · 2025-08-06
> >
> > Thanks for the clarification. After reading all reviews and rebuttals, I think the method has good contribution to mitigating high-frequency problem, and the method has advantage of bi-directional inference.

---

### Official Review · Reviewer_hdMN · 2025-06-30

**Clarity:** 2
**Significance:** 3
**Originality:** 3
**Rating:** 4
**Confidence:** 3

**Summary:**

The authors propose Global Fourier Modulation (GFM) to fix spectral bias in modulated INRs. They reparameterize network weights using fixed cosine bases (randomized in phase) + sample specific modulation coefficients.
They build upon the Functa idea, to represent paired function spaces (shapes, temporal fields,...) with shared latent vectors for bidirectional inference.

**Questions:**

+ How does GFM compare to other methods in terms of frequency metrics (powerspectrum, nmse,...)? If you bin low/high frequencies, how does it compare? Can you show empirical evidence for spectral bias mitigation?
+ Did you do a fair hyperparameter search for the other methods/architectures? In Table 1 the MSE of the KS-equation is over 7 for Shift; this seems unusually high for normalized data.
+ What is your main point you want your reader to take away?
    - GFM is good to mitigate spectral bias?
    - Exploration of shared latent modulation vector behaviour
    - Bidirectional inference possibilities
A clear focus on one topic helps the overall reading-flow. It was hard to follow in a first read. Currently, it is necessary to read into prior work to understand the methods presented.
+ Most of the used PDEs are toy datasets (Convection, Helmholtz), in regards to how much higher frequencies influence the PDE. Given that we know very little about the Reynoldsnumber of the datasets, it is hard to assume how much of a challenge the given PDEs are. Can you evaluate your method on dataset like Kolmogorov-Flow or different setups of the KS equation with set higher Reynoldsnumbers (so know to expect turbulences in the data)?
- Given the claim that your method can compete with NO and PINNs, there should also be more extensive experimental setups, that should be in the main paper (not appendix). There is a single table in the main paper (Table 4), which also indicated that the presented method is beatable by other methods. Can you show a more extensive anlysis of the PDEFuncta part of the paper? Compare on complex dataset the capabilities and to modern baselines.

**Ethical Concerns:**

["NO or VERY MINOR ethics concerns only"]

**Final Justification:**

The authors have been addressing some of my concerns via experiments/tables, however given that no images/plot could be shared within the rebuttal period, there are still open questions (comparison of powespectra).
In the believe that the authors will address the other issues I raised, in the camera-ready version I am going to reevaluate my given score.

**Limitations:**

yes

**Quality:**

2

**Strengths And Weaknesses:**

### Strengths
- Novel idea to enable bidirectional inference between paired function spaces via shared latents (meta-learning framework)
- Creative extension of the cosine reparametrization to modulate INRs
- Methods shows consistent improvements on their selected datasets
- Shows potential for unified forward/inverse operator learning

### Weaknesses
- Only a theoretical explanation for mitigation of spectral bias (no frequency domain analysis or frequency related metrics for empirical validation of claims).
- More extensive comparison to other spectral bias mitigation methods, to show the direct comparision on spectral metrics (SIREN, WIRE,...).
- Only limited scope of experiments shown, while claims are high (like solves spectral bias, the overall comparison to neural operators (NO) or physics informed neural networks (PINN) is weak from an experimental standpoint).
- Typing/Spelling mistakes in text/figures; Table and figure descriptions are weak/need more context
- Hard to read/follow/understand (missing connected overall story). Focusing on one topic (GFM or PDEFuncta) could have benefited the storyline.

The paper has interesting ideas, but validation of the spectral improvements are missing (spectral analysis), as well as the validation of beating most existing state of the art models. Experimental setups and results are mostly banished to the appendix. Also missing an extensive comparison to other spectral methods (SIREN,...).
The empirical improvements might exist trough the proposed method, but we do not know if they come from better spectral properties or just a better reparameterization in general.

---

> ### Author Rebuttal · Authors · 2025-07-31
>
> **W1 & Q1) How does GFM compare to other methods in terms of frequency metrics ... Can you show empirical evidence for spectral bias mitigation?**
>
> Thank you for the question. To empirically evaluate how GFM compares to other methods in the frequency domain, we conducted a Fourier RMSE analysis on the KS dataset, following prior approaches for assessing spectral bias.
> We computed the reconstruction error across low, mid, and high frequency bands and compared GFM to Shift, Scale, and FiLM-based modulation. The results are presented below:
> | Frequency Band | Shift | Scale | FiLM | Ours (GFM) |
> |----------------|-------|--------|------|------------|
> | **High**       | 6.396 | 4.335  | 3.432 | **1.386** |
> | **Mid**        | 0.011 | 0.015  | 0.016 | **0.055** |
> | **Low**        | 0.055 | 0.073  | 0.074 | **0.086** |
> | **Avg**        | 2.806 | 1.898  | 1.511 | **0.664** |
>
> As shown, GFM achieves the lowest average error and significantly improves accuracy in high-frequency bands, offering clear empirical evidence of spectral bias mitigation.
>
> ---
>
> **W2) More extensive comparison to other spectral bias mitigation methods, to show the direct comparision on spectral metrics (SIREN, WIRE,...).**
>
> Thank you very much for your valuable suggestion. As described in the supplementary source code, PDEfuncta utilizes SIREN as its backbone architecture. To provide a more extensive comparison, we have conducted additional experiments using alternative backbones, which are Wavelet Implicit Neural Representation (WIRE) [1] and Fourier Feature Mapping (FFM) [2]. The experiments were performed using the Convection dataset with coefficients $\beta$ ranging from 1 to 50.
>
> ### FFM Backbone
> | Modulation | PSNR | MSE |
> |------------|------|------|
> | Shift      | 21.127   | 0.0322  |
> | Scale      | 23.813 | 0.0175  |
> | FiLM       | 23.219 | 0.0186  |
> | **Ours (GFM)** | **29.879** | **0.0044** |
>
> ### WIRE Backbone
> | Modulation | PSNR | MSE |
> |------------|------|------|
> | Shift      | 27.492 | 0.0401 |
> | Scale      | 32.894 | 0.0029 |
> | FiLM       | 29.282 | 0.0047 |
> | **Ours (GFM)** | **33.047** | **0.0025** |
>
> Note that our proposed GFM modulation is theoretically grounded specifically in the characteristics of the SIREN backbone and hyperparameter settings are optimized accordingly. However, as clearly shown in the table, even with FFN and WIRE backbones, GFM consistently outperforms other modulation techniques. This underscores GFM's effectiveness in mitigating spectral bias across different architectures.
>
>
> *[1] Saragadam, Vishwanath, et al. "Wire: Wavelet implicit neural representations." Proceedings of the IEEE/CVF Conference on Computer Vision and Pattern Recognition. 2023.*
>
> *[2] Tancik, Matthew, et al. "Fourier features let networks learn high frequency functions in low dimensional domains." Advances in neural information processing systems 33 (2020): 7537-7547.*
>
> ---
>
> **W3 & Q5) Given the claim that your method can compete with NO and PINNs, there should also be more extensive experimental setups, … Can you show a more extensive anlysis of the PDEFuncta part of the paper? Compare on complex dataset the capabilities and to modern baselines.**
>
> We sincerely thank the reviewer for highlighting this important concern. The primary objective of our PDEfuncta framework is to accurately represent high-frequency content frequently observed in PDE solution snapshots. Our experiments confirmed that PDEfuncta successfully captures these complex high-frequency details, especially when modeling parametric PDE solutions with modulated INRs.
>
> We additionally demonstrated PDEfuncta's capabilities on downstream tasks such as neural operator scenarios, where it produced results comparable to modern neural operator baselines—achieving the best performance on the NACA dataset. However, we agree that to fully realize its potential as a general neural operator, further enhancements are necessary, and we will include this point explicitly in our discussion.
>
> Lastly, we would like to respectfully clarify that we have not claimed direct comparability to PINNs. We appreciate your pointing this out, and we will ensure this distinction is clearly communicated in our revision.
>
> ---
>
> **W4) Typing/Spelling mistakes in text/figures; Table and figure descriptions are weak/need more context**
>
> We acknowledge the importance of clear and detailed captions for tables and figures. Accordingly, we will significantly enhance their descriptions, adding more contextual information to aid comprehension and effectively highlight our results. These improvements will be reflected in the camera-ready version.
>
> ---
>
> **W5) Focusing on one topic (GFM or PDEFuncta) could have benefited the storyline.**
>
> We sincerely thank the reviewer for this valuable feedback. To clearly establish the connected storyline, we introduced Global Fourier Modulation (GFM) specifically as a modulation method addressing the spectral bias issue, which was an essential prerequisite step towards extending our framework to PDEfuncta. We acknowledge the reviewer's suggestion and will explicitly clarify in our revision why GFM is crucial and how it naturally leads into PDEfuncta.
>
> ---
>
> **Q2) Did you do a fair hyperparameter search for the other methods/architectures? In Table 1 the MSE of the KS-equation is over 7 for Shift; this seems unusually high for normalized data.**
>
> The Kuramoto–Sivashinsky (KS) dataset used in our experiments is not normalized, which naturally leads to relatively high MSE values. The raw solution fields contain large-magnitude, high-frequency variations due to the chaotic nature of the dynamics, so MSE values in the range of 7 are not unexpected in this setting.
>
> ---
>
> **Q3) What is your main point you want your reader to take away?**
>
> We sincerely appreciate the reviewer’s insightful and detailed comments. Below, we address each specific point:
>
> **Q3-1) GFM is good to mitigate spectral bias?**
>
> We fully agree that mitigating spectral bias is central to our study. As outlined in the manuscript, Global Fourier Modulation (GFM) directly addresses spectral bias, a prevalent limitation in standard implicit neural representations (INRs), by explicitly injecting frequency-aware priors into the modulation of INRs. The theoretical justification for this approach is provided clearly in Theorem 1, where Fourier reparameterization notably increases the gradient contributions for high-frequency components. Empirically, this improvement is demonstrated through rigorous experiments across various scientific PDE benchmarks (Figure 1, Figure 6, and Table 1), particularly highlighting GFM's superior capability to accurately capture high-frequency details compared to existing modulation methods.
>
> **Q3-2) Exploration of shared latent modulation vector behaviour**
>
> Indeed, in modulated INRs, latent vectors (z) typically serve as simple representations or compressed encodings of data. Our contribution extends this notion significantly by investigating the dynamics of these latent vectors in relation to PDE parameters. Specifically, in Appendix F, we provide detailed empirical analyses demonstrating that variations in latent vectors correspond closely to PDE coefficients. Further, we introduce PDEfuncta, which uniquely represents paired scientific datasets through a shared latent vector, thereby facilitating bidirectional inference within scientific machine learning scenarios.
>
> **Q3-3) Bidirectional inference possibilities ... it is necessary to read into prior work to understand the methods presented.**
>
> In our camera-ready version, we commit to enhancing the related work and proposed method sections to better contextualize PDEfuncta within the literature. Furthermore, Appendix I will be expanded to explicitly include comprehensive details about the inference process, thereby clearly illustrating the training-inference transition. Again, we are grateful for your valuable feedback, which will greatly enhance the readability and clarity of our manuscript.
>
>
> ---
>
> **Q4) Most of the used PDEs are toy datasets (Convection, Helmholtz), in regards to how much higher frequencies influence the PDE. ... Can you evaluate your method on dataset like Kolmogorov-Flow or different setups of the KS equation with set higher Reynoldsnumbers?**
>
> As demonstrated in prior work (e.g., [3]), approximating PDE solutions using neural networks presents its own set of challenges (e.g., spectral bias), even for problems that are traditionally considered tractable in numerical analysis. Examples include the convection equation with higher beta values and the Helmholtz equation with shorter wavelengths. Nevertheless, we greatly appreciate the reviewer’s suggestion and have conducted additional experiments to assess the applicability of the proposed method to more complex and challenging problem settings.
>
> To this end, we conducted an additional experiment using the Kuramoto–Sivashinsky (KS) equation, where we varied the viscosity parameter $\nu$ to induce different levels of turbulence. In our original setup, all training samples used a fixed viscosity $\nu = 0.1$. In the new experiment, we generated 100 training samples using five distinct viscosity values:
> $\nu$ $\in$ {0.05, 0.10, 0.15, 0.20, 0.25},$\quad$ $\text{20 samples per value}.$
> This setup results in a mixture of chaotic (low $\nu$) and smooth (high $\nu$) solutions, enabling a more comprehensive stress test.
>
> | Method | PSNR ↑ | MSE ↓ |
> |--------|--------|--------|
> | Shift  | 18.02  | 7.75   |
> | Scale  | 20.03  | 4.68   |
> | FiLM   | 22.02  | 3.21   |
> | **Ours** | **29.86**  | **0.61**   |
>
> Our model continued to perform robustly under this more complex and diverse setting, demonstrating its ability to generalize across multiple turbulence regimes.
>
>
> *[3] Krishnapriyan, Aditi, et al. "Characterizing possible failure modes in physics-informed neural networks." Advances in neural information processing systems 34 (2021): 26548-26560.*

---

> > ### Comment · Reviewer_hdMN · 2025-08-04
> >
> > Thank you for the extensive rebuttal.
> >
> > - W1: Thank you for the experiments to show what actually happens within certain spectral regions. The set boundaries to the frequency-bands have not been shown. I also assume that the produced trajectories are low resolution (32-256 pixels), which is hard to argue that there are higher-frequencies present.
> > However the given experiments show that for GFM it reconstructs only the higher frequencies better! If for example lower (or high energy) frequencies are more important for the task at hand, this method would perform worse than the basic shift operation. I am not sure that GFM overcome the issues of spectral bias, but it is a good method to mitigate some of the effects, with the trade of, to lose accuracy in lower frequency modes.
> >
> > - W2 + W3: Thank you, very clear setups, and explanations!
> >
> > - W4: Yes, please also visit your architectural plots as well, for example in Fig 3: $\gamma\left(\psi_u^L\left(W_u^L h_u^L+b_u^L\right) ; \alpha_u^L, \Phi_a^L\right)$ should be $\gamma\left(\psi_u^L\left(W_u^L h_u^L+b_u^L\right) ; \alpha_u^L, \Phi_u^L\right)$
> >
> >
> > - Q2: Is there any apparent reason to not normalize the data before you train your network? This is very odd, especially as it is common to do so if you use Scale/Shift/FiLM. This might make a larger difference for the other methods!
> >
> > - Q3: Thank you. I think this is very clear now to me; I hope the adopted version delivers the same content.
> >
> >
> > - Q4: It really depends on the used resolution of the PDE as well. For KS if you resolve different viscosity parameters at 2048x2048 and then downsize both to 64x64, they are basically identical in regard to their higher frequency content (as downsampling is like a low-pass filter). I think this experiment only make sense if you have high-resolution images, where you actually have different power-spectra the networks has to deal with.

---

> > > ### Author Response · Authors · 2025-08-05
> > >
> > > **W1/Q4 (Frequency Bands and Resolution)**
> > >
> > > We thank the reviewer for the insightful comments regarding both the frequency band boundaries and the resolution of the KS dataset.
> > >
> > > In our Fourier RMSE analysis, we applied FFT to the reconstructed and ground-truth solutions and then divided the resulting frequency spectrum into three equal-width bands—low, mid, and high—by uniformly partitioning the frequency axis into thirds. We acknowledge that this detail was not clearly stated above and will explicitly report the exact band boundaries and visualizations of the frequency spectrum in the camera-ready version. Additionally, we plan to explore task-adaptive strategies in future work to better align with the spectral characteristics of specific PDE problems.
> > >
> > > Regarding resolution, our current KS experiments use 51 time steps $\times$ 512 spatial grid points, which we will also explicitly state. This resolution enables a reasonably rich spectral representation. However, we fully agree that differences in spectral content—especially between turbulent and smooth regimes—become more pronounced at even higher resolutions. As the reviewer noted, downsampling acts as a low-pass filter and may mask meaningful high-frequency variations.
> > >
> > > To address this, we plan to conduct additional high-resolution experiments (e.g., 2048 points) to evaluate how resolution affects spectral bias mitigation and to analyze power spectra before and after reconstruction. We appreciate this suggestion, which will help us further assess and improve GFM’s performance in more challenging spectral settings.
> > >
> > > ---
> > > **W2+W3/Q3**
> > > We are glad to hear that the revised explanation helped clarify our contributions and we will ensure the same clarity is delivered in the adopted version.
> > >
> > > ---
> > > **Q2 (Normalization)**
> > > We thank the reviewer for this important observation. In our experiments, we followed the original setup established by [1], where the KS dataset is typically used without normalization to preserve its natural scale and inherently chaotic dynamics. This allowed for direct comparison across modulation methods under consistent conditions. We acknowledge that normalization is common in some settings and may influence the relative performance of modulation techniques. We plan to include additional experiments with normalized KS data in future work and will clarify this design choice in the camera-ready version.
> > >
> > > [1] Ruiz, Ricardo Buitrago, et al. "On the benefits of memory for modeling time-dependent PDEs." arXiv preprint arXiv:2409.02313 (2024).
> > >
> > > ---
> > > **W4 (Figures)**
> > >
> > > We will thoroughly revise Figure 3 and all architectural plots to correct formatting issues and improve clarity.
> > >
> > > ---
> > > Again, thank you for your constructive feedback and encouragement—it has been invaluable in refining our work.

---

### Official Review · Reviewer_CbRV · 2025-07-03

**Clarity:** 3
**Significance:** 3
**Originality:** 3
**Rating:** 4
**Confidence:** 2

**Summary:**

The authors introduce Global Fourier Modulation (GFM) which addresses the challenge of spectral bias in implicit neural representations (INRs) when modeling complex solution fields of partial differential equations (PDEs). They further propose PDEfuncta, a meta-learning framework that learns bidirectional mappings between paired function spaces.  Experiments demonstrate substantial improvements in reconstruction accuracy and generalization to unseen parameters and resolutions without retraining.

**Questions:**

* Neural Operators have been successfully applied to spatiotemporal evolution applications. The applications in the paper seem to be all steady state examples. Can PDEFuncta be extended to spatiotemporal evolutions?
* Can the authors provide frequency power spectrum plots of the GFM reconstructions and report the fourier error metrics[1] for the evaluated methods?

[1] Appendix B in "PDEBench: An Extensive Benchmark for Scientific Machine Learning", Takamoto et al., NeurIPS 22

**Ethical Concerns:**

["NO or VERY MINOR ethics concerns only"]

**Final Justification:**

Good paper for mitigating spectral bias in operator learning. Finds use in many physics problems with high frequency information like turbulence modeling and fine-scale phenomena like vortices.

**Limitations:**

Yes

**Quality:**

3

**Strengths And Weaknesses:**

### Strengths:
* Bidirectional Operator Learning: PDEfuncta supports both forward and inverse mappings in a unified latent space, matching or exceeding neural operator baselines which gives it a clear advantage over PINN and Neural Operators
* GFM seems to provide a huge gain in terms of PSNR as compared to Shift, Scale or FiLM
### Weaknesses:
* The mitigation of spectral bias has not been shown empirically (e.g. with the help of frequency power spectrum plots)

---

> ### Author Rebuttal · Authors · 2025-07-31
>
> **Q1) The mitigation of spectral bias has not been shown empirically (e.g. with the help of frequency power spectrum plots). Can the authors provide frequency power spectrum plots of the GFM reconstructions and report the fourier error metrics[1] for the evaluated methods?**
>
> **A1)** We thank the reviewers for highlighting the importance of explicitly evaluating spectral bias. To address this, we performed an additional frequency-domain analysis on the Kuramoto–Sivashinsky (KS) dataset using Fourier RMSE, which measures the root-mean-square error between the ground-truth and reconstructed solutions in the frequency domain. Specifically, we computed the error within three frequency bands: low, mid, and high.
>
> The results are summarized below:
>
> | Method | High ↓ | Mid ↓ | Low ↓ | Avg ↓ |
> |--------|--------|--------|--------|--------|
> | Shift  | 6.396  | 0.011  | 0.055  | 2.806  |
> | Scale  | 4.335  | 0.015  | 0.073  | 1.898  |
> | FiLM   | 3.432  | 0.016  | 0.074  | 1.511  |
> | Ours   | 1.386  | 0.055  | 0.086  | 0.664  |
>
> [Table 1] Fourier RMSE across frequency bands on KS dataset
>
> These results confirm that our GFM approach significantly reduces reconstruction error in the high-frequency regime compared to baseline modulation methods. While the low and mid-frequency components are reconstructed comparably across methods, GFM exhibits clear advantages in preserving high-frequency content, directly addressing the concern of spectral bias.
>
> -----
>
> **Q2) Neural Operators have been successfully applied to spatiotemporal evolution applications. The applications in the paper seem to be all steady state examples. Can PDEFuncta be extended to spatiotemporal evolutions?**
>
>
> **A2)** We thank the reviewer for this observation. PDEfuncta is readily extendable to spatio-temporal evolution tasks. In our Navier–Stokes experiment (see Figure 3), we model a temporal operator by treating two contiguous time windows as paired function spaces:
>
> $\mathcal{A}$: $t \in [0, T_a]$ is represented by an INR $f_a$,
>
> $\mathcal{U}$: $t \in [T_a, T_u]$ is represented by a second INR $f_u$
>
> Both networks share the same latent vector $z$. Consequently, the latent map $G : \mathcal{A} \rightarrow \mathcal{U}$ learns the flow operator that advances the solution by $\Delta t=T_u−T_a$. This procedure can be applied to long-horizon roll-outs on the Navier-Stokes dataset and other dynamical benchmarks.

---

### Official Review · Reviewer_5L3v · 2025-07-04

**Clarity:** 3
**Significance:** 3
**Originality:** 3
**Rating:** 5
**Confidence:** 3

**Summary:**

This paper proposes to combine FiLM-like conditioning with Fourier-based weight construction to achieve latent representations that are transferable across dataset samples, which are subsequently integrated in a Deep-O-Net like model to predict PDE solutions, called implicit neural representation (INR). For each data sample, a latent vector is constructed, which is used to produce scaling and bias factors to modulate networks weights. Based on that, the authors further extend to use the same latent weights to decode both the input and output of a PDE mapping. Experiments are conducted on diverse 1-step mapping PDE problems.

**Questions:**

- From Algorithm 1 in Appendix I, it seems that no extra model is used to compute the latent vector z (such as an auto-encoder), is this correct? Is this the standard way for training INRs?

- Would the proposed method be extensible to PDEs without periodic boundaries?

Typo:

- Line 171: the first $\alpha^{k.W}$ should use comma between $k$ and $W$.

**Ethical Concerns:**

["NO or VERY MINOR ethics concerns only"]

**Final Justification:**

The authors clarified my concerns. However, my knowledge of INRs restricts me from providing a higher rating. Thus, I will maintain my rating as accept.

**Limitations:**

- Limitation is discussed in Appendix B.

**Paper Formatting Concerns:**

No major formatting issue is noticed.

**Quality:**

3

**Strengths And Weaknesses:**

Strengths:

- The claims that GFM creates better high-frequency reconstructions seem to be supported by experiments.

- The proposed method outperforms baseline models, including Shift, Scale, and FiLM.

Weekness:

- Based on my understanding, while GFM exploits the structure of Fourier bases, which filters over space and time for both global and local patters, the baseline methods (Scale, Shift, FiLM) do not. For example, from the bottom row of Figure 6 it seems that the baseline models barely work for the demonstrated sample. As multi-resolution processing is known to be critical to model PDEs [1], a stronger baseline model would also make use of some global-local processing, such as convolutions at multiple scales.

- The latent interpolation experiment in 5.2.1 appears to be rather easy as the tested beta values are quite dense within the training set betas.

[1] Towards Multi-spatiotemporal-scale Generalized PDE Modeling, https://arxiv.org/abs/2209.15616

---

> ### Author Rebuttal · Authors · 2025-07-31
>
> **Q1) Based on my understanding, while GFM exploits the structure of Fourier bases, which filters over space and time for both global and local patterns, the baseline methods (Scale, Shift, FiLM) do not. For example, from the bottom row of Figure 6 it seems that the baseline models barely work for the demonstrated sample. As multi-resolution processing is known to be critical to model PDEs [1], a stronger baseline model would also make use of some global-local processing, such as convolutions at multiple scales.**
>
> Thank you for your constructive suggestion! While we acknowledge that the suggested method could serve as a valuable baseline for comparison, adapting it to the INR framework may be nontrivial. We note that Figure 6 illustrates the reconstruction results of INRs with different modulation techniques, not neural operators results. We will expand the discussion on this aspect—particularly regarding local and global feature representations—by citing the referenced work in the revised manuscript.
>
> ---
>
> **Q2) The latent interpolation experiment in 5.2.1 appears to be rather easy as the tested beta values are quite dense within the training set betas.**
>
> We thank the reviewer for pointing this out. The default interpolation setting in Section 5.2.1 indeed uses densely sampled training parameters (interval = 1), which could make interpolation relatively easier.
>
> To more rigorously evaluate PDEfuncta's generalization ability, we designed additional interpolation experiments, where interpolation evaluation is performed under increasingly challenging conditions. Specifically, during test time, we restrict the available latent codes to those corresponding to training $\beta$ values sampled at coarser intervals (e.g., 2, 3, 5), and perform interpolation only using those. This allows us to assess the model’s ability to interpolate under limited latent support.
>
> | Interval | PSNR ↑ | MSE ↓   |
> |----------|--------|---------|
> | 1        | 32.59  | 0.0017  |
> | 2        | 26.90  | 0.0057  |
> | 3        | 21.29  | 0.0230  |
> | 4        | 14.91  | 0.1035  |
> | 5        | 11.75  | 0.2593  |
>
> [Table 1] Latent interpolation under reduced support
>
> While the performance gradually degrades as the available latent support becomes sparser, the model still produces reasonable reconstructions. Notably, even at interval 5, PDEfuncta outperforms Shift, Scale, and FiLM baselines evaluated under the easier interval-1 setting (see Figure 5 and Table 2 in the main paper). This further supports the robustness of our learned latent space and its ability to generalize under limited information.
>
> ---
>
> **Q3) From Algorithm 1 in Appendix I, it seems that no extra model is used to compute the latent vector z (such as an auto-encoder), is this correct? Is this the standard way for training INRs?**
>
> Yes, the reviewer is correct. As shown in Algorithm 1, our method does not use an explicit encoder (such as in a VAE or autoencoder) to compute the latent vector \( z \). Instead, we adopt an *auto-decoder* approach, where each z is treated as a learnable parameter initialized randomly and optimized directly via gradient descent.
>
> This approach is widely used in implicit neural representation frameworks such as DeepSDF [1] and Functa [2]. It provides more direct control over per-instance representations and is particularly suitable for high-fidelity reconstruction tasks as in our PDE scenarios.
>
> [1] Park, Jeong Joon, et al. "Deepsdf: Learning continuous signed distance functions for shape representation." Proceedings of the IEEE/CVF conference on computer vision and pattern recognition. 2019.
>
> [2] Dupont, Emilien, et al. "From data to functa: Your data point is a function and you can treat it like one." International Conference on Machine Learning (pp. 5350-5364). (2022).
>
> ---
>
> **Q4) Would the proposed method be extensible to PDEs without periodic boundaries?**
>
> Thank you for the insightful question. We would like to clarify that the proposed method does not require periodic boundary conditions. While the term "Fourier" appears in the GFM design, it refers to a reparameterization of the MLP weights in a frequency-aware space, and does not assume periodicity of the input domain.
>
> Importantly, several of our evaluated benchmarks involve non-periodic boundaries. For instance, both the Airfoil and Pipe datasets include non-periodic boundaries, yet PDEfuncta achieves strong performance on these tasks (see Table 4). These results demonstrate the method's ability to generalize beyond non-periodic boundary settings.
>
> ---
>
> **Q5) [Typo issue] Line 171**
>
> We thank the reviewer for pointing this out. As noted, the correct notation should be $\alpha^{k, W}$ instead of $\alpha^{k.W}$. We will fix this typo in the revised version.

---

### Official Review · Reviewer_FpZz · 2025-07-07

**Clarity:** 3
**Significance:** 3
**Originality:** 3
**Rating:** 4
**Confidence:** 4

**Summary:**

In this work, the authors propose a novel modulation technique for Implicit Neural Representations (INR), called Global Fourier Modulation (GFM) that injects high-frequency information at each layer of the INR through Fourier-based reparameterization. This enables compact and accurate representation of multiple solution fields using low-dimensional latent vectors. Moreover, the authors adopt GFM to define PDEfuncta, a meta-learning framework designed to learn multi-modal PDE solution fields and support generalization to new tasks. The authors demonstrate that the method not only improves representational quality but also shows potential for forward and inverse inference
tasks without the need for retraining.

**Questions:**

Why did the authors focused on a discrete set of parameters to test PDEfuncta?
How can one identify the correct latent space? What is the suggested size of it and how does it affect PDEfuncta?

**Ethical Concerns:**

["NO or VERY MINOR ethics concerns only"]

**Final Justification:**

Considering the authors' rebuttal to my initial comments, the main question I raised were answered. I am still a little doubtful about the choice of the beta values, because for larger intervals, the solution is quite degraded (at least in terms of MSE). I feel that the choice of these beta values is crucial for the correct training and usage of PDEfuncta, and that there is room for improvement in how to effectively sample beta values to achieve superior performances. Therefore, I maintain my initial ratings.

**Limitations:**

The manuscript lacks benchmarks with broadband solution to be approximated. The experiments presented in the manuscript are relatively low frequency and it would be interesting to perform a stress-test on broader-band exercises

**Quality:**

3

**Strengths And Weaknesses:**

PDEfuncta shows the highest PSNR and lowest MSE across all datasets for 1D benchmarks. The major weakness relies on the auto-encoding of the input data to identify the latent space. This could represents a bottleneck depending on the chosen technique. While optimal performance may require careful hyperparameter tuning to select the appropriate frequency ratio, this flexibility enables GFM
framework to be adapted to a wide range of scientific problems and data characteristics.

---

> ### Author Rebuttal · Authors · 2025-07-31
>
> **Q1) The major weakness relies on the auto-encoding of the input data to identify the latent space. This could represents a bottleneck depending on the chosen technique.**
>
> Thank you for the insightful comment. An alternative approach to circumvent the latent bottleneck is to train separate INRs individually, for example, one per $\beta$ value when solving the convection equation. While this can enhance expressivity, it incurs a substantial computational cost. This trade-off has also been widely discussed in the computer vision community, motivating approaches like Functa, which offer significantly better efficiency while still delivering comparable performance. A central challenge for these methods, however, lies in improving their expressivity, which our work aims to address.
>
> ---
>
> **Q2) Why did the authors focused on a discrete set of parameters to test PDEfuncta?**
>
> In the context of parameterized PDEs, solution snapshots—which we aim to represent using INRs—can vary significantly with different values of the input parameters, such as $\beta$ in the convection equation. For instance, lower beta values produce smoother, less oscillatory solutions, while higher values lead to highly oscillatory ones. Drawing an analogy to computer vision tasks, we can think of a solution snapshot at a given beta as an image, and assume that there exists a collection of such images corresponding to a discrete set of beta values. Therefore, our use of a discrete parameter set follows standard practices in PDE benchmark datasets and allows for systematic training and evaluation.
>
> While our model is trained on discrete $\beta$ values, we emphasize that PDEfuncta is not limited to memorizing those specific samples. To assess its ability to generalize to unseen, continuous parameter values, we conducted an additional interpolation experiment under the same setup as Section 5.2.1 (Setting #1). Specifically, we increased the spacing of the training $\beta$ values to intervals of 2 and 3 (from the default of 1), and kept the test $\beta$ values fixed at the midpoints (e.g., $\beta$ = 1.5, 2.5, ...). This setting makes interpolation more challenging by forcing the model to infer from sparser training support. Results are as follows:
>
> | Interval | PSNR ↑  | MSE ↓ |
> |--------|-----------|----------|
> | 1 (default) | 32.59 | 0.0017 |
> | 2 | 26.90 | 0.0057 |
> | 3 | 21.29 | 0.0230 |
>
> [Table 1] Interpolation performance with increasing gaps between training $\beta$ values.
>
> These results support our claim that PDEfuncta learns a smooth and structured latent space that generalizes beyond the discrete training set, capturing functional variations across continuous parameter ranges.
>
> ---
>
> **Q3) How can one identify the correct latent space?**
>
> Evaluating the "correctness" of a learned latent space is challenging, as there is no ground-truth latent code for comparison. To address this, we designed diagnostic experiments (cf. Section 5.2.1) to assess the functional behavior of the latent space. In particular, we consider the latent space meaningful if it satisfies the following two properties:
> Latent interpolation: Interpolating between latent codes of nearby training parameters yields accurate predictions for intermediate, unseen parameter values.
> Partial generalization: A latent code inferred from partial observations should enable reconstruction of the full solution for unseen parameter instances, without updating the shared INR.
> These two capabilities indicate that the latent space captures smooth and structured functional variation aligned with the underlying PDE family.
>
> ---
>
> **Q4) What is the suggested size of it and how does it affect PDEfuncta?**
>
> We thank the reviewer for raising this important point. In the main paper, we fixed the latent code dimension to 20 across all tasks for simplicity. To evaluate the sensitivity of PDEfuncta to the latent dimensionality, we conducted additional experiments with smaller (10) and larger (40) latent codes.
> The table below summarizes the results for Helmholtz #1 (uni-directional) and Navier-Stokes (bi-directional):
> | Code Size | PSNR (Helmholtz) ↑ | PSNR (NS-A) ↑ | PSNR (NS-U) ↑ |
> | --------- | ------------------ | ------------- | ------------- |
> | 10 | 38.42 | 42.70 | 43.15 |
> | 40 | 39.75 | 43.79 | 43.78 |
>
> These results indicate that PDEfuncta is robust to the choice of latent code dimension. Even with a small code size (i.e., 10), the model performs competitively, and increasing the dimension provides only modest gains. This shows that PDEfuncta captures functional variation efficiently without requiring large latent capacity.
>
> ---
>
> **Q5) The manuscript lacks benchmarks with broadband solution to be approximated. The experiments presented in the manuscript are relatively low frequency and it would be interesting to perform a stress-test on broader-band exercises**
>
> We thank the reviewer for the constructive feedback. While the experiments in the paper include datasets with nontrivial frequency content—such as the Kuramoto–Sivashinsky (KS) equation, which is widely used to benchmark high-frequency behaviors in PDE learning—we agree that broader-band scenarios offer a stronger stress test for evaluating spectral generalization.
>
> To further support our claims, we conducted two additional experiments:
>
> 1. KS with varying Reynolds numbers: We varied the viscosity ($ν$ $\in$ $\[0.5, 0.6, ..., 3.0\]$) to change the Reynolds number and produce solutions with different turbulence levels and spectral densities.
> 2. Convection with extended $\beta$ range (1–100): By sampling $\beta$ from a wider range, we induce sharper gradients and more oscillatory behavior in the solution fields.
>
> | Method | PSNR ↑ | MSE ↓ |
> |--------|--------|--------|
> | Shift  | 18.02  | 7.754  |
> | Scale  | 20.03  | 4.682  |
> | FiLM   | 22.02  | 3.215  |
> | Ours   | 29.86  | 0.613  |
>
> [Table 2] Reconstruction performance on KS with varying Reynolds number
>
> | Method | PSNR ↑ | MSE ↓ |
> |--------|--------|--------|
> | Shift  | 14.82  | 0.342  |
> | Scale  | 22.42  | 0.159  |
> | FiLM   | 23.69  | 0.086  |
> | Ours   | 35.65  | 0.0013 |
>
> [Table 3] Reconstruction performance on Convection with $\beta$ $\in$ $\[1, 100\]$
>
> These results further validate that PDEfuncta can handle broadband signal regimes and remains effective under severe spectral challenges.

---

> > ### Comment · Reviewer_FpZz · 2025-08-08
> >
> > I would love to thank the reviewers for taking the time to carefully reply to my questions. I am still concerned with the choice of the beta values, since it seems that the model is highly sensitive to the sampling strategy adopted. I therefore maintain my rating.

---

### Decision · Program_Chairs · 2025-09-17

**Decision:**

Accept (poster)

**Comment:**

The paper addresses the spectral bias problem in implicit neural representations (INRs) for learning to represent high-frequency components often found in physical fields. The application domain is learning PDE solutions, with the objective of obtaining compressed functional representations of physical fields characteristic of a given phenomenon.

The authors propose a new technique, Global Fourier Modulation (GFM), which directly parameterizes INR weights using shared cosine bases for a given PDE, along with PDE-sample-specific modulation coefficients. GFM is designed to modulate frequency information in the network weights directly, enabling finer control than previous methods. The idea is further instantiated for learning neural operators between two function spaces, based on a shared latent representation, allowing the method to learn paired representation spaces. Experimental validation is conducted on a series of reconstruction and prediction problems.

The reviewers highlight the novelty of the proposed reparameterization, describing it as a solid and original technical contribution that achieves improvements over baselines. In their initial reviews, they noted some weaknesses, mainly concerning the experimental validation. During the rebuttal, the authors addressed most concerns and added new experiments to respond to the reviewers’ comments. The overall consensus is clearly in favor of acceptance.